# Methane- and dissolved organic carbon-fueled microbial loop supports a tropical subterranean estuary ecosystem

D. Brankovits [1], J.W. Pohlman [1,2], H. Niemann [3,4,10], M.B. Leigh [5], M.C. Leewis [5,6], K.W. Becker [7], T.M. Iliffe[1], F. Alvarez[8], M.F. Lehmann [3] & B. Phillips[9]

Subterranean estuaries extend inland into density-stratified coastal carbonate aquifers containing a surprising diversity of endemic animals (mostly crustaceans) within a highly oligotrophic habitat. How complex ecosystems (termed anchialine) thrive in this globally distributed, cryptic environment is poorly understood. Here, we demonstrate that a microbial loop shuttles methane and dissolved organic carbon (DOC) to higher trophic levels of the anchialine food web in the Yucatan Peninsula (Mexico). Methane and DOC production and consumption within the coastal groundwater correspond with a microbial community capable of methanotrophy, heterotrophy, and chemoautotrophy, based on characterization by 16S rRNA gene amplicon sequencing and respiratory quinone composition. Fatty acid and bulk stable carbon isotope values of cave-adapted shrimp suggest that carbon from methanotrophic bacteria comprises 21% of their diet, on average. These findings reveal a heretofore unrecognized subterranean methane sink and contribute to our understanding of the carbon cycle and ecosystem function of karst subterranean estuaries.

---

[1] Department of Marine Biology, Texas A&M University at Galveston, Galveston, TX 77553, USA. [2] U.S. Geological Survey, Woods Hole Coastal and Marine Science Center, Woods Hole, MA 02543, USA. [3] Department of Environmental Sciences, University of Basel, Basel, 4056, Switzerland. [4] Department of Marine Microbiology and Biogeochemistry, NIOZ Royal Netherlands Institute for Sea Research, 1790 AB Den Burg, Netherlands. [5] Institute of Arctic Biology, University of Alaska Fairbanks, Fairbanks, AK 99775, USA. [6] U.S. Geological Survey, GMEG Science Center, Menlo Park, CA 94025, USA. [7] Department of Marine Chemistry and Geochemistry, Woods Hole Oceanographic Institution, Woods Hole, MA 02543, USA. [8] Institute of Biology, National Autonomous University of Mexico (UNAM), 04510 Mexico, D.F., Mexico. [9] Speleotech, Tulum, 77780 Quintana Roo, Mexico. [10] Centre for Arctic Gas Hydrate (CAGE), 9037 Tromsø, Norway. Correspondence and requests for materials should be addressed to D.B. (email: david.brankovits@gmail.com) or to J.W.P. (email: jpohlman@usgs.gov)

Recognition that chemosynthetic biological communities capture energy and synthesize organic matter (OM) from chemicals emitted from the seafloor[1,2] reshaped our understanding of life on Earth[3] and the oceanic carbon cycle[4]. OM produced by chemosynthetic pathways and expelled from seafloor hydrothermal vents and cold seeps may exceed 10% of surface ocean productivity[4,5]. Submarine groundwater discharge is another important source of nutrients and carbon to the ocean[6], in some instances exceeding that of rivers[7], and therefore represents a critical exchange vector between continental land-masses and the ocean[8]. However, much more is known about the magnitude of submarine groundwater discharge to the coastal ocean than the genesis of the material expelled[7,9]. Evidence that a coastal aquifer food web in the Yucatan peninsula (YP) (Mexico) is partially dependent on a chemosynthetic food source[10] suggests mutual biogeochemical reactions govern the ecology of these Earth–Ocean transition zones, yet the basic carbon cycle in this widely distributed coastal aquifer ecosystem remains largely unexplored. In the present study, we adapt methods previously used to investigate continental-margin cold seeps[11–13] to delineate the biogeochemistry and functional ecology of this coastal aquifer.

Mixing of terrestrial meteoric water with saline groundwater (SGW) in coastal aquifers resembles the two-layered circulation of surface estuaries such that they have been termed subterranean estuaries[14]. Subterranean estuaries are found globally along siliciclastic, basaltic, and karstic (carbonate) coastlines[7,9,15]. The most prevalent and human-accessible estuary type is found within porous limestone of karst coastlines, where marine-derived groundwater extends inland beneath the meteoric lens flooding extensive cave passages[16,17]. Karst coastlines account for ~25% of coastlines globally[15] and ~12% of all submarine groundwater discharge[9]. Research conducted by scientific divers within cave conduits of coastal aquifers has led to a basic understanding of stygobitic (cave-limited) macrofaunal biodiversity[18,19] within this globally distributed ecosystem (termed anchialine, meaning near the sea)[20], the food web structure[10], how sea level change during the Holocene affected the habitat development[21], and hydrologic controls that govern the mixing of fresh and marine waters within the subterranean estuary[17].

The seminal investigation of anchialine ecosystems suggested OM supporting consumers of the food web in a tropical subterranean estuary was partially derived from a chemoautotrophic source[10]. This conclusion was based on the bulk stable carbon isotopic composition of several crustacean species that were distinct ([13]C-depleted) from available photosynthetic sources and similar to invertebrates from deep sea vent communities that rely on a chemoautotrophic food base. Comparable isotopic values were reported for invertebrates from a thermomineral cave in Romania with clear evidence that mantle-derived hydrogen sulfide ($H_2S$) was the primary energetic source[22]. The anchialine ecosystem investigated by Pohlman et al.[10] contained no $H_2S$ or other evidences of mantle derived material, suggesting that non-sulfurous reduced compounds (e.g., ammonium or methane) liberated during OM decomposition support microbial communities. However, they were unable to definitively constrain the nutritive OM source. Subsequent studies from a freshwater karst aquifer[23], a sunlit anchialine sinkhole[24], an alluvial aquifer[25], and freshwater lakes[26] also suggest higher trophic level invertebrates utilize chemoautotrophic products generated from OM degradation, supporting the possibility that ecosystems deep within a coastal aquifer are sustained by similar processes.

In this study, we investigated the carbon cycle and food web dynamics of a pristine anchialine ecosystem within a tropical karst subterranean estuary in Mexico's YP. The YP is a limestone platform that contains more than 1000 km of mapped cave conduits within the coastal region of the Holbox fracture zone[27] (Fig. 1). These cave passages prevail within the inland portion of the subterranean estuary over an area (~1100 km²) comparable to surface estuaries like Galveston Bay (Texas) (~1500 km²), the 7th largest estuary in the U.S. Natural sinkholes, locally known as cenotes, provide scientists direct access to the flooded caves. The site we investigated (Cenote Bang) is located ~8 km inland within a mature dry tropical forest, and is one of the entrances to the Ox Bel Ha cave network (Fig. 1c; Supplementary Fig. 1).

Based on the observation that a complex food web exists in coastal karst aquifers with limited particulate OM, we tested the hypothesis that dissolved organic carbon (DOC)—including methane—formed from decomposition of terrestrial vegetation within water saturated limestone beneath the tropical forest provides carbon and energy for a microbial loop that, in turn, supports the subterranean food web[28]. We identified carbon sources and inferred biogeochemical cycles based on the distribution, concentration, and isotopic composition of organic and inorganic carbon compounds, and electron acceptor availability. We characterized the microbial community by sequencing of 16S ribosomal RNA (rRNA) genes and identifying quinone lipid biomarkers from environmental water samples. To link the microbes to the food web, we performed compound-specific isotopic analysis of membrane-derived fatty acids (FAs) extracted from filter-feeding cave-adapted shrimp. This multifaceted study provides a broad perspective for carbon transformations and exchange between the terrestrial and marine realms of a tropical karst subterranean estuary.

## Results

**Water column properties**. To characterize the physical and chemical environment of caves accessed from Cenote Bang (Fig. 1), we collected sonde profiles during four sampling campaigns (Fig. 2) between 2013 and 2016 in the Ox Bel Ha Cave System. For all events, we observed three distinct water masses separated by thin (20–60 cm) haloclines (H1 and H2) that were relatively constant in depth (Fig. 2). Salinity in the layer nearest the cave ceiling of the shallowest passages (~3 m water depth) ranged from 0.3 to 0.7 psu, which was slightly less than the cenote pool (0.9–1.8 psu). Salinity ranged from 2.0 to 2.5 psu in the middle layer, and from 34.8 to 37.6 psu in the deepest layer. Sampling of the deep SGW was restricted to 22 m depth below groundwater table due to the geometry of the cave passages. To differentiate the subterranean water masses, we hereafter refer to the low salinity water mass as meteoric freshwater (MFW), the intermediate salinity water mass as meteoric brackish water (MBW), and the deep water layer underlying the meteoric lens as SGW. Moreover, we refer to the coastal sea water as SEA and the open-to-air sinkhole/cenote as POOL, recognizing that the POOL is part of the meteoric lens (Fig. 1e).

Dissolved oxygen in the MFW was at or near anoxia (0–15 μM) and constant in the vertical extent for each campaign. The SGW displayed the highest dissolved oxygen (DO) content (45–55 μM) (Fig. 2), but was still always hypoxic (<60 μM). The MBW showed two distinct profile types. During August 2014 and January 2015, DO was mostly invariant with depth in the MBW (22–29 μM). By contrast, during December 2013 and January 2016, MBW was anoxic near the shallow halocline (H1) and increased gradually with increasing depth toward the deeper halocline (H2). During the days preceding the sonde profilings, there was substantially more rainfall in December 2013 (457 mm) and January 2016 (253 mm) than during August 2014 (52 mm) and January 2015 (39 mm) (Supplementary Table 1). DO in the POOL was consistently low (10–37 μM), but always elevated relative to the MFW and MBW during each event.

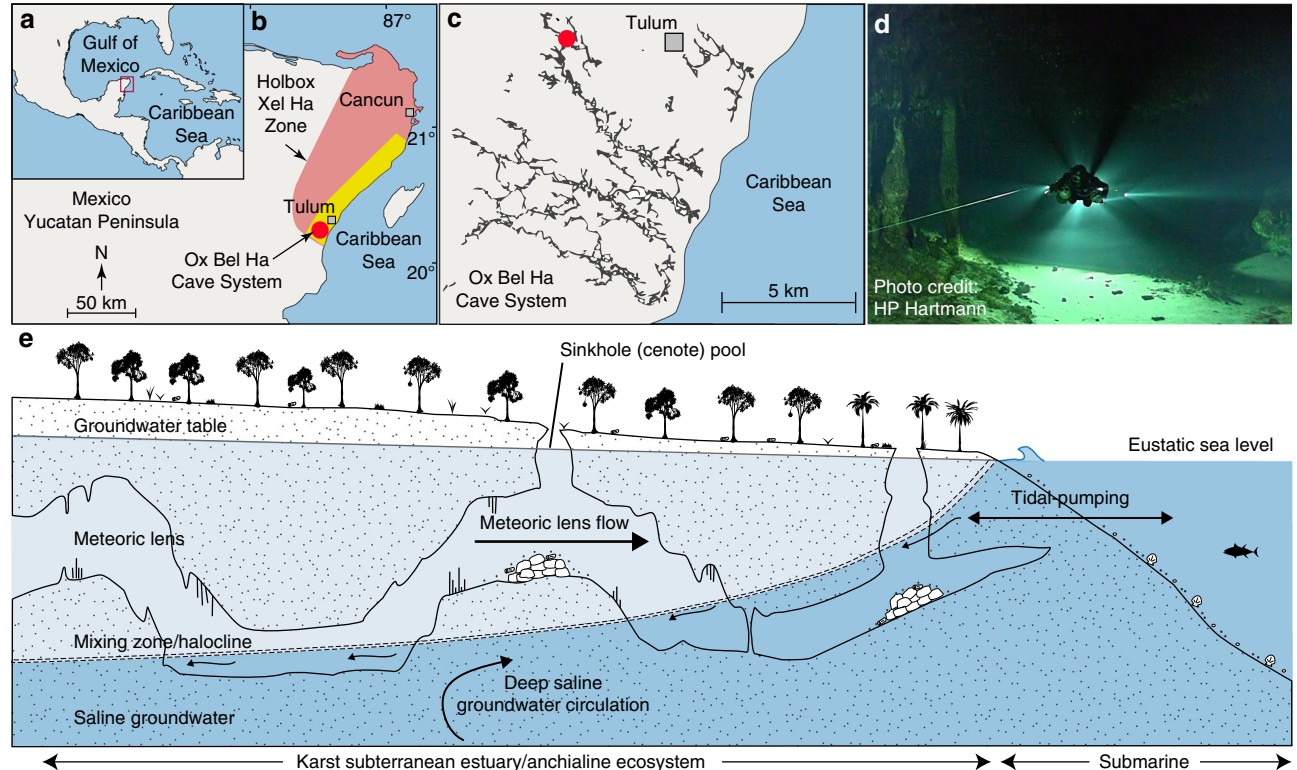

**Fig. 1** Study site and model for coastal karst subterranean estuary. **a** The Yucatan Peninsula. **b** In the Yucatan Peninsula (Mexico), coastal cave systems extend 12 km inland (yellow area) and cover an area of ~1100 km² within the Holbox Fracture Zone and Xel Ha Zone (red area)—adapted from Perry et al.[16] **c** Mapped cave conduits of Ox Bel Ha Cave System (>240 km total length)[27]. Cenote Bang, the primary study site, is indicated by the red circle. **d** Cave diver within cave system Cenote Bang. **e** Conceptual model of a tropical karst subterranean estuary, a density-stratified coastal aquifer—adapted from van Hengstum et al.[21]

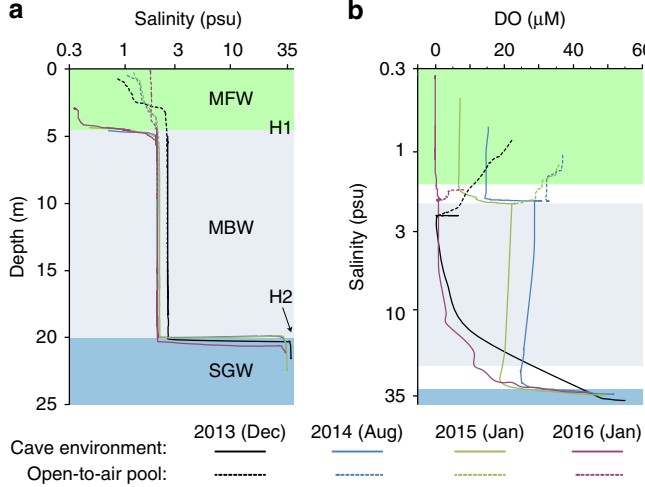

**Fig. 2** Physicochemical profiles from the karst subterranean estuary. **a** Salinity-depth profiles. **b** Dissolved oxygen-salinity profiles. Continuous lines are cave profiles, and dashed lines are open-to-air cenote pool (POOL) profiles. H1 (halocline 1) separates meteoric freshwater (MFW) from the meteoric brackish water (MBW), and H2 (halocline 2) separates the MBW from the saline groundwater (SGW)

**Aqueous biogeochemistry**. To investigate the distribution, sources, and turnover of dissolved and particulate carbon in the water masses, we collected 63 samples from the three water layers and analyzed them for concentration and δ¹³C values of dissolved inorganic carbon (DIC), particulate organic carbon (POC), DOC,

and methane ($CH_4$), as well as sulfate and chloride concentrations (Supplementary Data 1). Table 1 provides average values for each parameter measured during the course of the study as a baseline for characteristic concentrations and carbon isotopic contents for each water mass, including the open-to-air cenote pool and the coastal ocean (additional details in Supplementary Tables 2, 3).

The anoxic-MFW had the highest methane concentrations (3550–9522 nM) with $\delta^{13}C$-$CH_4$ values (−66.3 ± 0.7‰; Fig. 3a, b; Supplementary Fig. 2) that are characteristic of microbial methane[29]. Methane concentrations in the hypoxic-MBW (43–275 nM) were lower than those from the POOL (100–890 nM), and about an order of magnitude less than the MFW. MBW (−52.7 ± 1.9‰) and POOL (−50.6 ± 4.9‰) $\delta^{13}C$-$CH_4$ values were similar to each other, but substantially more ¹³C-enriched than observed for the MFW. The $\delta^{13}C$-$CH_4$ values from December 2013, following a period of exceptional precipitation (Supplementary Table 1), were the most ¹³C-enriched. The hypoxic-SGW had the lowest methane concentrations in the aquifer (37–208 nM) and were similar to the coastal sea values (43–235 nM). The $\delta^{13}C$-$CH_4$ values in the SGW (−56.3 ± 1.5‰) were comparable to those in nearby coastal ocean waters (−59.0 ± 2.1‰) and were ¹³C-enriched relative to the MFW. Compared to the concentration and carbon isotopic ranges predicted from conservative (non-reactive) mixing models that use MFW and deep SGW methane end member values (Fig. 3a, b), the intermediate depth MBW $CH_4$ concentrations were lower and $\delta^{13}C$-$CH_4$ values were higher, indicating methane removal by oxidation[29].

Like methane, DOC concentrations were highest in the anoxic-MFW (402–834 μM), an order of magnitude lower in the MBW (37–203 μM), and lowest in the deep SGW (15–80 μM; Fig. 3c, d;

**Table 1 Aqueous biogeochemistry**

| | Meteoric freshwater MFW | Meteoric brackish water MBW | Saline groundwater SGW | Sinkhole (cenote) POOL | Coastal water SEA |
|---|---|---|---|---|---|
| Salinity (psu) | 0.26 ± 0.03 (8) | 1.81 ± 0.04 (29) | 32.87 ± 0.94 (13) | 0.94 ± 0.09 (6) | 35.45 ± 0.39 (6) |
| $[SO_4^{2-}]$ mM | 0.3 ± 0.1 (7) | 1.6 ± 0.1 (27) | 26.4 ± 1.0 (11) | 0.8 ± 0.1 (6) | 28.4 ± 0.6 (4) |
| $[CH_4]$ nM | 6466 ± 659 (8) | 157 ± 16 (28) | 110 ± 17 (12) | 495 ± 148 (6) | 121 ± 28 (6) |
| $\delta^{13}C\text{-}CH_4$ ‰ | −66.3 ± 0.7 (7) | −52.7 ± 1.9 (25) | −56.3 ± 1.5 (11) | −50.6 ± 4.9 (6) | −59.0 ± 2.1 (5) |
| [DOC] µM | 661 ± 132 (3) | 131 ± 16 (16) | 41 ± 20 (3) | – | – |
| $\delta^{13}C\text{-}DOC$ ‰ | −28.0 ± 0.1 (3) | −28.3 ± 0.2 (16) | −26.6 ± 0.4 (3) | – | – |
| [DIC] mM | 4.4 ± 0.2 (6) | 7.1 ± 0.2 (24) | 2.4 ± 0.2 (10) | 5.3 ± 0.2 (3) | 2.0 ± 0.1 (3) |
| $\delta^{13}C\text{-}DIC$ ‰ | −16.4 ± 1.0 (7) | −11.1 ± 0.7 (25) | −6.3 ± 1.0 (11) | −9.4 ± 2.1 (3) | −4.3 (2) |
| POC µM | 10.9 ± 3.8 (3) | 5.0 ± 2.3 (3) | 3.0 ± 0.9 (3) | 32.3 ± 14.9 (3) | 5.8 (1) |
| $\delta^{13}C\text{-}POC$ ‰ | −28.5 ± 0.5 (3) | −27.6 ± 0.7 (3) | −27.1 ± 1.0 (3) | −28.0 ± 0.3 (3) | −20.1 (1) |

Values of water column constituents, presented as average ± std. error (n), from the different regimes of the groundwater system and the adjacent coastal sea. Values were calculated from all measurements within a water mass across all sampling events. For further information regarding data obtained during the separate sampling events, see the supplement (Supplementary Tables 2, 3)

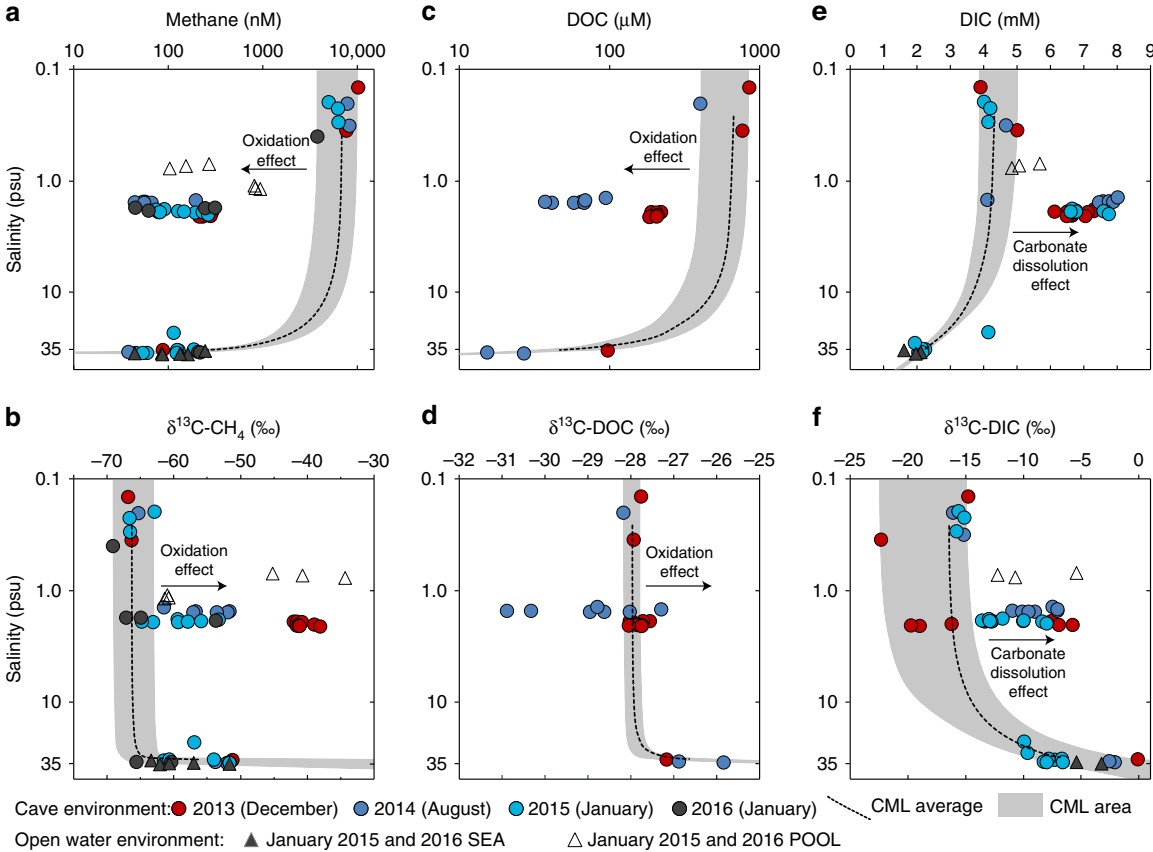

**Fig. 3** Plots of salinity vs. chemical properties from the subterranean estuary. **a** Methane concentrations. **b** Dissolved organic carbon (DOC) concentrations. **c** Dissolved inorganic carbon (DIC) concentrations. **d** Methane stable carbon isotopic ($\delta^{13}C$) values. **e** DOC stable carbon isotopic ($\delta^{13}C$) values. **f** DIC stable carbon isotopic ($\delta^{13}C$) values. The average and total area of conservative mixing lines (CMLs; see Methods for calculations) represent the trend predicted by the mixing model if there was only physical mixing between the meteoric freshwater and saline groundwater end members. Production yields an excess of the constituent relative to the CML average and area, while consumption results in depletion. Symbol of individual data points contain the uncertainty (1 std. dev.) of the measured values

Supplementary Fig. 2). The MFW $\delta^{13}C$-DOC values (−28.0 ± 0.1‰) were consistent with a terrestrial C3 plant origin[30], the dominant vegetation in the overlying tropical forest. The SGW $\delta^{13}C$-DOC values (−26.6 ± 0.4‰) were slightly higher due to contributions from the coastal ocean[31]. Similar to methane, DOC concentrations in the MBW were much lower when compared to predictions from the conservative mixing model, indicating removal of DOC (Fig. 3c, d). However, for August 2014, the majority of the DOC samples displayed low $\delta^{13}C$ values (Fig. 3d),

opposite of the effect expected for oxidation. Consistent with the distribution of DOC, the highest POC concentrations occurred in the anoxic-MFW (3.3–14.6 µM). However, POC does not contribute significantly to the total organic carbon pool, with average concentrations only 1.6% of the DOC. The $\delta^{13}C$-POC values in the MFW (−28.5 ± 0.5‰) are consistent with a forest vegetation origin[30].

DIC was the largest pool of carbon in the cave waters (Fig. 3e, f; Supplementary Fig. 2). Biological respiration, carbonate

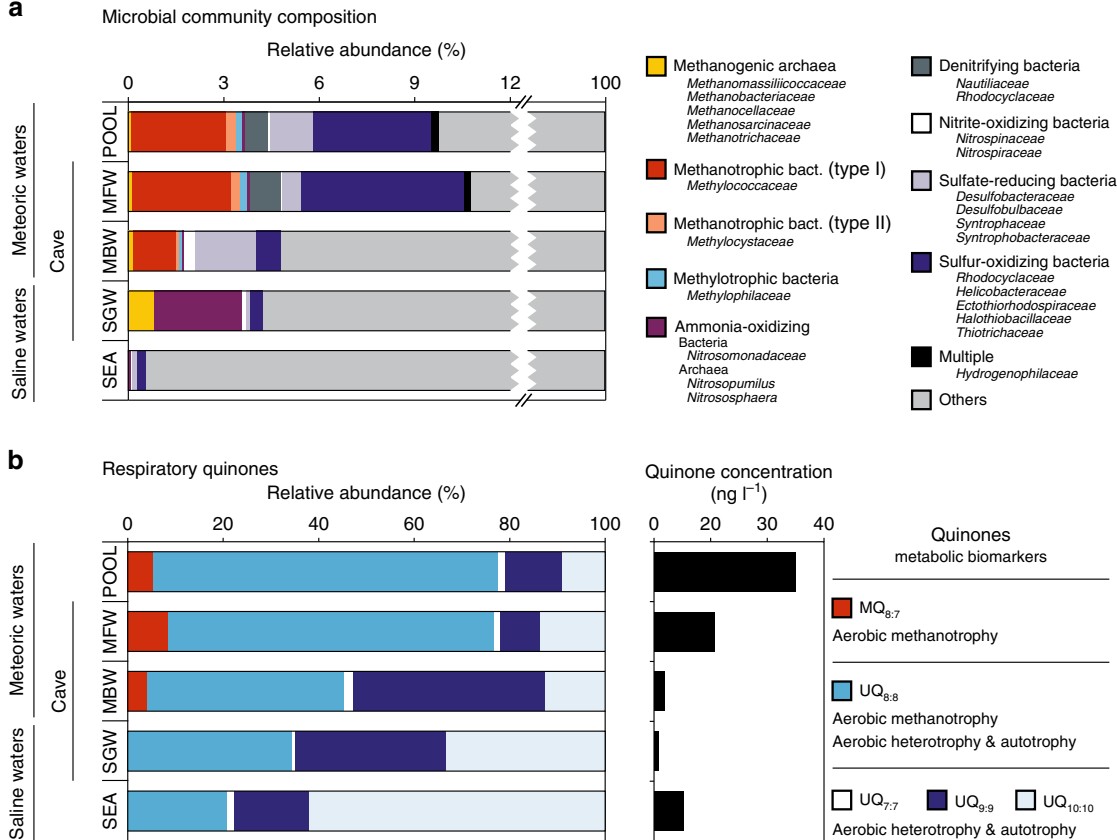

**Fig. 4** Microbial community diversity within the subterranean and surface water regimes. **a** 16S rRNA amplicon sequence community analysis. Functional groupings are based on the presence of sequences shared by microbes that mediate the associated function. "Multiple" indicates there was no specific function associated with the closest match type strain. The majority of sequences were from bacteria and archaea capable of utilizing a variety of complex organic compounds ("Others"; see Supplementary Fig. 4 for further details). For a comprehensive phylogenetic listing and relative abundance of sequence reads see Supplementary Data 2. **b** Respiratory quinones relative abundance, concentrations, and metabolic affiliations. Quinone nomenclature ($UQ_{m:n}$) after Elling et al.[35], where Q indicates headgroup type, $m$ number of isoprenoid units in the side chain, and $n$ number of double bonds. MBW, meteoric brackish water; MFW, meteoric freshwater; POOL, cenote pool; SEA, coastal sea; SGW, saline groundwater

dissolution, and chemolithotrophic $CO_2$ assimilation have the potential to alter DIC concentration and carbon isotopic ratios. High concentrations of DIC in the MBW require the addition of DIC from a $^{13}C$-enriched source. The most likely source for such a large input of DIC is dissolution of carbonate, which occurs within this groundwater mixing zone[32] and has $\delta^{13}C$ values that are ~0‰[33]. Sulfate, a potential electron acceptor for OM respiration and/or the anaerobic oxidation of methane, did not vary in concentration relative to the conservative mixing diagram (Supplementary Fig. 3).

**Microbial community**. To characterize the microbial community structure in the density-stratified aquifer, the open-water cenotes, and the coastal ocean, we sequenced 16S rRNA genes and analyzed respiratory quinone biomarkers from January 2016 water samples (Fig. 4). Phylogenetic affiliations were assigned based on 16S rRNA gene sequences and were clustered into metabolic groups by inferred phenotypes of representative sequences within each operational taxonomical unit. These functional groups were determined to identify microbes capable of mediating biogeochemical pathways inferred from geochemical analyses (Fig. 4a). Because the water samples were collected within the water masses and not at the interfaces between water masses (where we hypothesize carbon consumption to be most active), the sequence data are a qualitative indicator of the microbial community composition. Nevertheless, the cenote pool, MFW, and MBW

showed relatively high abundances of sequences from type I (1.2–2.3%) and type II (<0.3%) methanotrophic bacteria, sulfur-oxidizing bacteria, and other archaeal and bacterial functional groups involved in methylotrophy, as well as chemoautotrophic nitrogen and sulfur cycling processes. The relative abundance of methanogenic archaea was below 1%, with highest abundance in the SGW, where ammonia-oxidizing microbes were also present (Fig. 4). Numerous other microbes capable of utilizing a wide range of organic compounds were identified ("Others" in Fig. 4a; Supplementary Fig. 4; Supplementary Data 2).

Quinone biomarkers offer DNA-independent detection and quantification of microbial biomass in samples from the natural environment[34, 35]. In our samples, the occurrence and relative distribution of quinones were distinct for the meteoric and saline water regimes (Fig. 4b). The major quinone types in all samples were ubiquinones (UQs) containing 7–10 isoprenoid units and 1 double bond per isoprenoid unit (see quinone nomenclature in Methods). Additionally, in the samples from the POOL, the MFW and MBW methylene-ubiquinone ($MQ_{8:7}$) was detected, which structurally differs from regular UQs by the presence of a methylene group in the isoprenoid side chain. In the samples from the POOL and MFW, $UQ_{8:8}$ was the dominant quinone (72% relative abundance), while in the samples from the MBW, $UQ_{8:8}$ and $UQ_{9:9}$ contribute 40% and 41% to total quinones, respectively. $UQ_{8:8}$, $UQ_{9:9}$, and $UQ_{10:10}$ were equally distributed in the deep SGW, while $UQ_{10:10}$ was the dominant quinone with

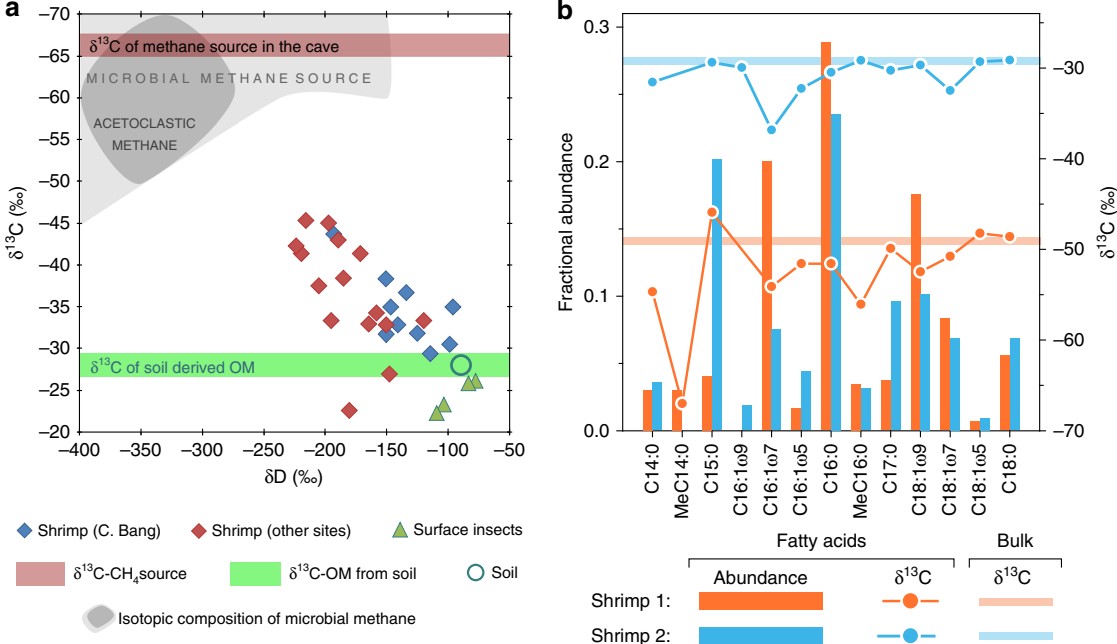

**Fig. 5** Stable isotopic content of food sources and consumers. **a** Carbon and deuterium stable isotope ratios ($^{13}C/^{12}C$, $^2H/^1H$) from stygobitic shrimp (diamonds) and surface insects (triangles) plotted relative to $CH_4$ and soil-derived organic matter (OM) end members. Proximity to source indicates relative trophic dependency. The $\delta^{13}C$ value of methane from the MFW ($-66.3 \pm 0.7$‰, red shaded area) is consistent with a microbial source[29] (gray shaded area). The $\delta^{13}C$ of the soil-derived OM ($-28.0 \pm 0.2$‰, green shaded area) includes values for forest soil, cave POC, and cave DOC. **b** Fatty acids from stygobitic shrimp specimens. Methane-derived carbon contribution is higher (55%) for shrimp 1 (orange) and lower (3%) for shrimp 2 (blue) based on bulk $\delta^{13}C$ values ($-49.1$‰ and $-29.3$‰, respectively). The combination of extremely negative $\delta^{13}C$ values and bacterial origins for odd-numbered monounsaturated and methylated FAs indicates a bacterial food source that is likely to include aerobic methanotrophs. Symbols of individual data points contain the uncertainty (std. dev.) of the measured isotopic values

62% in the coastal sea water. Highest quinone concentrations occurred in the POOL with 31 ng l$^{-1}$. In the cave, the concentrations decreased with increasing salinity across the different water regimes (MFW, MBW, SGW) (Fig. 4b). The quinone $MQ_{8:7}$ is diagnostic for type II methanotrophs and $UQ_{8:8}$ has been shown to be the dominant quinone in type I methanotrophs[36], but the latter is also widespread among other aerobic bacteria[34]. The SGW and coastal ocean (SEA) samples, in contrast, mainly contained $UQ_{9:9}$ and $UQ_{10:10}$, which occur in diverse aerobic bacteria.

**Bulk stable isotopes**. To determine the trophic relationships between potential food sources and consumers, specimens ($n = 29$) of stygobitic *Typhlatya* spp., a free-swimming atyid shrimp with feeding appendages capable of capturing bacteria-sized particles[10, 37] and insects ($n = 4$) from the surface jungle, were measured for stable carbon and deuterium isotopic content (Fig. 5a). Twelve shrimp were obtained from the MBW of Cenote Bang cave. The remaining specimens ($n = 17$) were collected from the MBW and SGW of three locations connected (via cave conduits) with the main research site and two caves at greater distance that are not likely linked to Cenote Bang (Supplementary Fig. 5). Shrimp stable carbon isotope values ranged from −22.5 to −49.1‰, and the stable hydrogen isotope values ranged from −95.7 to −223.6‰ (Fig. 5a). These isotope values were between the terrestrial soil/insect values and those expected for microbial methane[29]. The measured $\delta^{13}C$-$CH_4$ values from the cave were typical for microbial methane, and the $\delta^{13}C$-DOC values were similar to those of the soil OM.

**Fatty acid biomarkers**. We report fractional abundances and stable carbon isotope values of FAs extracted from two shrimp

specimens (collected from Cenote Bang) with relatively small (3%) and large (55%) contributions of methane carbon to the specimen's biomass, as calculated from a two-source mixing model[25, 38]. We observed a range of $C_{14}$-$C_{18}$ FAs, all of which displayed $\delta^{13}C$ values similar to the specimen's bulk $\delta^{13}C$ (Fig. 5b; Supplementary Table 4). FA compounds extracted from the shrimp with relatively high methane contribution to its biomass (Shrimp 1; Fig. 5b) displayed more negative $\delta^{13}C$ values than FAs from the tissue of Shrimp 2 (Fig. 5b). Both shrimp contained generic, saturated FAs with an even number of carbon atoms ($C_{14:0}$, $C_{16:0}$, and $C_{18:0}$), as well as odd number unsaturated and methylated lipid compounds.

## Discussion

The results presented above demonstrate that methane and DOC derived from degraded terrestrial OM are the primary carbon and energy sources for a karst subterranean estuary ecosystem beneath an undisturbed tropical forest (Fig. 6). Variability in the DO profiles (Fig. 2b) and carbon chemistry (Fig. 3) of the water column suggests external factors influence the spatial and temporal dynamics of the aquifer biogeochemistry. However, the emphasis of this study and the following discussion is to identify unifying characteristics for developing a generic model of ecosystem function for this terrestrially influenced subterranean estuary to be applied to other anchialine ecosystems.

The most basic physical characteristic for this coastal aquifer and others[17, 24] is the uniform and extreme density stratification of the 25 m water column. The three distinct water masses separated by two sharp haloclines were present in the cave conduits during all sampling campaigns (Fig. 2a). The physicochemical characteristics of the MFW in the cave were distinct from the POOL, which had slightly higher salinity (~ 1.0 psu) and

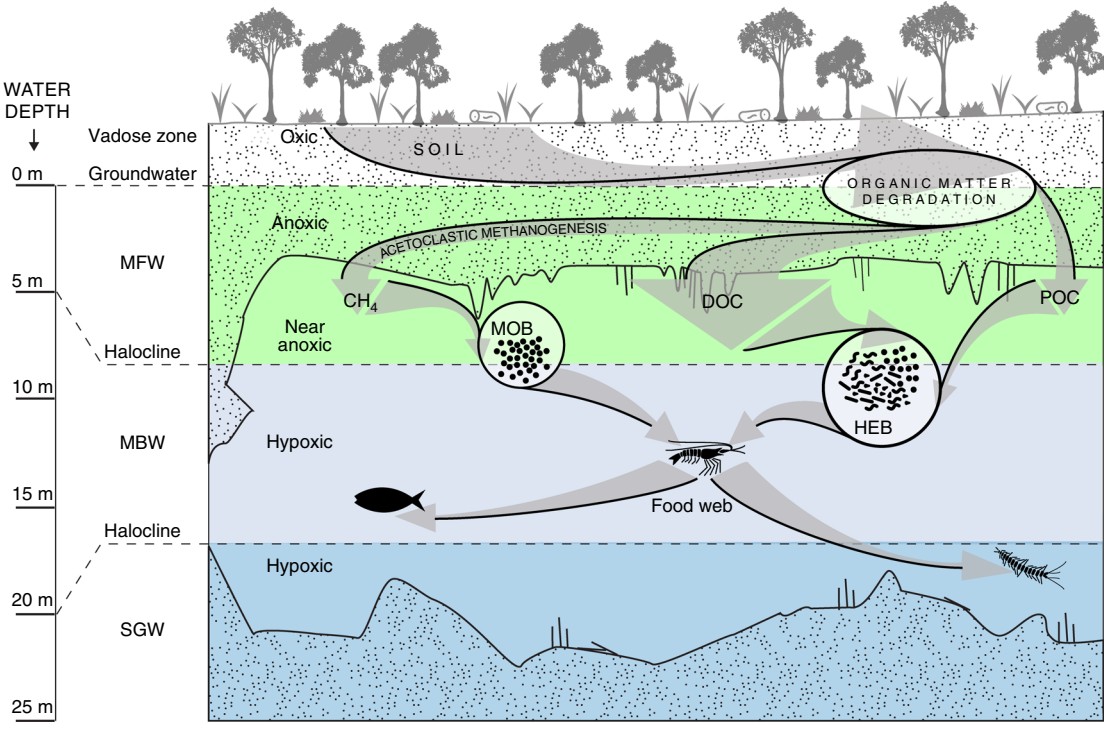

**Fig. 6** Conceptual model for a terrestrially influenced tropical karst subterranean estuary microbial loop. Dissolved organic carbon (DOC) and methane ($CH_4$) produced from soil organic matter degradation within the shallow and anoxic saturated zone of the carbonate rock-matrix are transported into hypoxic cave conduits, where methane oxidizing bacteria (MOB) and heterotrophic bacteria (HEB) consume these reduced forms of organic carbon. Bacterial biomass is assimilated by filter-feeding crustaceans that are, in turn, preyed upon by higher trophic levels of the food web in this anchialine ecosystem

oxygen (10–37 µM) contents. MFW occurs throughout the permeable karst aquifer (Fig. 1e); however, access to that portion of the aquifer was restricted to shallow, domed cave passages that extend vertically upward to water depths of 5 m or less. Herein, we argue the MFW is of critical importance to the carbon cycle of terrestrially influenced habitats in the anchialine ecosystem.

The shallowest portion of the aquifer is in contact with soil OM within saturated fissures and pores of the carbonate rock[15]. Anaerobic decomposition of this soil OM from the overlying tropical jungle is the most likely source of the high concentrations of DOC ($665 \pm 132$ µM) and methane ($6466 \pm 659$ nM) measured in the cave-accessible MFW. The stable carbon isotopic composition of the DOC (−28.0‰) is consistent with an origin from C3 vegetation of the overlying tropical forest[30]. An exponential relationship ($r^2 = 0.87$) between DOC and methane concentration suggests that methane and DOC originate from a similar source (Supplementary Fig. 6); one that likely involves syntrophic interactions between fermentative bacteria and methanogenic archaea. The average MFW $\delta^{13}$C-$CH_4$ value (−66.3‰) is consistent with a microbial methane origin by chemoautotrophic $CO_2$ reduction or acetoclastic methanogenesis[29]. Limited but detectable abundance of 16S rRNA gene sequences from methanogenic archaea in the MFW (Fig. 4) suggests that methane production takes place elsewhere. This further supports the hypothesis that the saturated zone within the permeable rock matrix[15, 16], a portion of the aquifer that was inaccessible to divers, was the most likely source of DOC in the MFW.

The juxtaposition of OM-charged, near-anoxic MFW against OM-poor, hypoxic-MBW is analogous to redox transitions present near sediment–water interfaces[39], the chemocline of meromictic lakes[40] and oxygen-minimum zones in the ocean[41], with the important exception that the relative positioning of the water masses within this coastal aquifer is inverted, or "upside-down", relative to open-water systems. High-OM, low-oxygen regimes in

sediments, and the water column of lakes and oceans are driven by pulses of sinking particulate detritus produced in surface waters or within the watershed basin. The accumulation and consumption of organic detritus depletes oxygen below the chemocline. For OM oxidation to continue, oxygen or alternate electron acceptors (e.g., sulfate, nitrate, etc.) must be replenished by mixing. By contrast, in the tropical karst aquifer we investigated, depleted oxygen (Fig. 2b) co-occurred with concentrated methane (Fig. 3a) and DOC (Fig. 3c) above the shallow chemocline (H1). The relatively high concentrations of oxygen in the deepest sampled portion of the aquifer (SGW) is consistent with the transport of DO with sea water moving inland from the coast below the deeper halocline[17] (H2 in Fig. 2). Distinct DO profiles preceded by periods of high and low rainfall (Fig. 2b) suggest precipitation is the key external factor regulating electron acceptor availability in the meteoric portion of the aquifer. We hypothesize that rainfall injects oxygenated water into the MBW at discrete entrances by point recharge, and drives DOC-enriched water from the anoxic saturated portion of the aquifer (the MFW) into the caves by diffuse recharge[15].

Previous studies in caves suggest POC concentrations are limited in karst groundwater[42, 43]. To evaluate POC bioavailability and origin in this coastal karst aquifer, we measured concentrations and $\delta^{13}$C values of POC for June 2015 and January 2016 (Supplementary Table 3) and compared them to DOC concentrations and $\delta^{13}$C values in the cave environment. Like DOC, POC is most abundant in the MFW (10.9 µM) and derived from the tropical forest vegetation, as indicated by its stable carbon isotopic composition ($\delta^{13}$C = −28.5‰). However, on average, DOC in the MFW is 60 times more abundant than POC. By comparison, DOC:POC ratios range between 6 and 10 in the surface ocean, rivers, and streams[44]. In the oligotrophic Atlantic Ocean, where DOC:POC ratios from 300 m water depth[31] are comparable to the MFW, DOC is the primary source of carbon

available to a microbial loop that supports the pelagic ecosystem[45]. We conclude, as others have for cave streams[43], some riverine systems[46] and oligotrophic oceans[45], that DOC is a more important source of carbon and energy than POC for this coastal aquifer ecosystem.

Carbon-based concentration and isotopic mixing models (Fig. 3) provide insight into evaluating if and where different carbon stocks are created or consumed within a mixing system. This approach has been used to investigate carbon dynamics in estuaries[47] and high-salinity marine pore waters[11]. We applied this concept to this subterranean estuary, and calculate using equation (4) that, on average, 93% of the methane (Fig. 3a) and 76% of the DOC (Fig. 3c) were removed within the MBW. The methane concentration reduction was 6300 nM and the DOC reduction was 530 μM, on average, suggesting there is an active sink for methane and bulk DOC. During oxidation of methane and DOC, $CO_2$ is certainly produced, however, production of $^{13}C$-enriched DIC in the MBW resulting from carbonate dissolution[32] (Fig. 3e, f) overwhelms the isotopic effect from the respired $CO_2$.

Enrichment of $^{13}C$ in the methane within the MBW is consistent with microbial oxidation being the removal mechanism. During enzymatic oxidation of methane, there is a bias toward utilization of the $^{12}C$-isotope, leaving the residual methane $^{13}C$-enriched[29], as observed here. By contrast, although the concentration-based mixing model for DOC indicates removal (Fig. 3c), a large positive carbon isotopic shift was not observed for the December 2014 data (Fig. 3d). This observation does not, however, conflict with the model evidence because isotopic fractionation of DOC during aerobic oxidation[48] is less than what occurs during methane oxidation[29]. The negative shift in the $\delta^{13}C$ of DOC for the August 2014 sampling event suggests production of DOC from methane carbon[13]. Conservative mixing of sulfate during all sampling events (Supplementary Fig. 3) indicates that sulfate reduction did not considerably contribute to the dissolved OM oxidation, but this analysis may not be sufficiently sensitive to detect changes of sulfate relative to carbon pools with orders of magnitude lower concentrations. The presence of DO in the MBW is additional evidence that methane and DOC oxidation were aerobic.

Analyses of microbial community structure (16S rRNA) and respiratory quinones reveal a diverse microbial community with distinct structuring within the karst subterranean estuary, the open-water cenote and the coastal ocean (Fig. 4; Supplementary Fig. 4; Supplementary Data 2). Sequences representing microbes that consume methane, utilize sulfur-and nitrogen-based electron acceptors to oxidize OM, as well as sulfur-oxidizers were relatively abundant in the freshwater portion of the aquifer (MFW) and sinkhole (POOL). In particular, type I methanotroph sequences from the genus *Methlyococcacea* were present in the POOL, as well as the MFW and MBW water masses, where geochemical data clearly indicate methane oxidation (Fig. 3).

The presence of respiratory quinones, which are lipid-soluble components of the electron transport chain[35, 49], provide evidence for metabolically active microbial functional groups in the subterranean estuary. The most prevalent quinones are affiliated with aerobic heterotrophic bacteria (Fig. 4b), which is consistent with the metabolic capacity of most microbes observed ("Others" in Fig. 4a) and with DOC being the most abundant form of OM consumed. In the meteoric water masses, the predominant quinone was $UQ_{8:8}$, which only occurs in strictly aerobic and facultatively anaerobic (grown under aerobic conditions) organisms and is the dominant quinone in type I methanotroph cultures[36]. This compound peaked in abundance in the MFW and POOL locations, where we also found highest 16S rRNA gene copy numbers of type I methanotrophs (Fig. 4). Detectable

concentration of the quinone biomarker $MQ_{8:7}$, which has only been found in type II methanotrophs[36], was also present in the portion of the groundwater where methane was oxidized. The dominance of type I over type II methanotrophs is not surprising, because they are generally more prevalent in environments with low oxygen[50], like those observed in this subterranean estuary, and are more efficient at converting methane carbon to biomass than are type II methanotrophs[51].

Sequences from numerous genera that mediate chemoautotrophic carbon fixation and utilization through oxidation and reduction of sulfur- and nitrogen-based compounds were also present in the open-air cenote and cave (Fig. 4). However, we presently have no evidence that these microbes contribute to the carbon cycle or food web of the caves we investigated. A sulfate mixing model similar to the carbon mixing models (Fig. 3) did not indicate removal of sulfate in the cave (Supplementary Fig. 3). Furthermore, none of the passages we investigated contained detectable $H_2S$. By contrast, deep open-water cenotes found in the YP, where organic debris accumulates near the deeper halocline (H2 in Fig. 2a) are most certainly settings where the carbon and sulfur cycles are intertwined[52]. Microbes from those areas may have been transported into the interior of the YP limestone platform. Alternatively, a cryptic sulfur cycle is active[41] or the mixing model lacks the sensitivity required to detect changes in sulfate concentration. Nitrification within the mixing zone of the MBW and SGW has been suggested as another potential chemoautotrophic source of OM in a YP anchialine ecosystem[10]. Near this interface, we found the coexistence of ammonia oxidizers typically found in either marine (*Nitrosopumilus*) or terrestrial environments (*Nitrososphaera*) (Fig. 4). However, given the relatively low concentrations of nitrate (18.6 μM) accumulated near the MBW–SGW interface[10] relative to the amount of DOC consumed (530 μM), and the low carbon assimilation efficiency of nitrifying bacteria, the likelihood that nitrification contributes meaningful nutritive carbon to the food web remains speculative[10]. Nevertheless, the sequence data are consistent with the hypothesis that multiple biogeochemical cycles utilizing all available electron donors and acceptors are active in these oligotrophic and anoxic/hypoxic habitats[28]. Additional studies are required to evaluate their importance for the food web. Our data support that DOC (including methane) derived from decomposition of terrestrial OM is the prevalent source of nutritive carbon that sustains the ecosystem.

Bulk stable carbon and hydrogen isotopic data from *Typhlatya* spp. shrimp adapted to feed on bacteria-sized, suspended matter in the water column[10] are consistent with a mixed dietary dependence on methane- and DOC-derived carbon (Fig. 5a) via the consumption of microbial biomass. Considering the range of shrimp tissue $\delta^{13}C$ values (−23 to −49‰), and the average MFW $\delta^{13}C$ values of methane (−66.3‰) and DOC (−28.0‰) (Table 1) as potential end members of the shrimp's dietary carbon source, the contribution of methane carbon for the shrimp ranges from 0 to 55%, with an average contribution of 21% (Supplementary Table 5). Studies from a humic lake[53] and an alluvial aquifer[25] report methane carbon contributions to zooplankton and insects ranging from 5 to 67%. Because the low δD values in the shrimp are distinctive for methane carbon incorporation[54], we used the shrimp bulk isotope values to estimate the δD signature of the methane source. By extrapolating the carbon and deuterium stable isotope values from the cave shrimp to the average MFW $\delta^{13}C$-$CH_4$ source value (−66.3‰), we estimated the δD-$CH_4$ signature was about −390‰, which allowed us to constrain that the microbial methane was produced by acetoclastic methanogenesis[29].

Methane and DOC-derived carbon flow into the anchialine food web is facilitated by trophic interactions between the shrimp

and its microbial food source. To examine the biochemical origins of the $^{13}C$-depleted values of the bulk shrimp tissue, we analyzed the composition and carbon isotopic content of FA biomarkers extracted from shrimp with the largest (55%) and smallest (3%) calculated methane contributions (Fig. 5b; Supplementary Table 4). The composition and $\delta^{13}C$ of FAs differed substantially between the two shrimp, which is consistent with assimilation of different food and carbon sources. In addition, in both specimens, individual FAs differed in their stable carbon isotopic composition, which is indicative of different biochemical pathways for FA synthesis[12]. Even-chained saturated FAs ($C_{14:0}$, $C_{16:0}$, and $C_{18:0}$) had compound-specific $\delta^{13}C$ values closely matching the bulk tissue (Fig. 5b), suggesting the shrimp synthesized these compounds de novo from their dietary carbon sources. In contrast, several monounsaturated FAs in both shrimp and methylated FAs in shrimp 1, showed substantially more negative $\delta^{13}C$ values than the bulk tissue (Fig. 5b), which provides evidence for a dietary source of these compounds through the direct transfer of FAs from ingested bacteria[12, 55]. Given the strong geochemical evidence for aerobic methane oxidation (Fig. 3) in the hypoxic environment, we attribute the source of $^{13}C$-depletion in the FAs primarily to methane carbon derived from methanotrophs. Stable isotope probing experiments using Arctic lake sediments have shown that several compounds present in the shrimp tissue ($C_{14:0}$, $C_{16:1\omega7}$ and $C_{18:1\omega7}$) are synthesized from methane carbon by methanotrophs that were also present in the cave waters (e.g., *Methylococcaceae* and *Methylocystacea*)[51]. These FAs and the methylated FAs are also found in other organisms, but are not likely to have the $^{13}C$-depleted isotope signatures observed here. A symbiotic source[56] of the incorporated bacterial biomass might also explain the observed FA profiles. However, *Typhlatya* shrimp appendages capable of direct filter-feeding of bacteria-sized particles[10] suggest the methanotrophic biomass is incorporated from the environment rather than from symbiotic sources. These observations have significant implications for understanding energy transfer within the anchialine food web. Considering that *Typhlatya* spp. are one of the most abundant macrofaunal populations in this habitat[19], and they are the primary prey for predators in the subterranean food web[57], it is reasonable to assume they have a significant role in transferring methane- and DOC-derived carbon to higher levels of the food web.

The geochemical, genomic, and biomarker evidence from this study supports the hypothesis that a microbial loop is active in a karst subterranean estuary ecosystem (Fig. 6). We demonstrated that DOC and methane produced from soil OM degradation within the shallow, anoxic saturated zone of the karst are transported downwards into hypoxic cave conduits, where methanotrophs and heterotrophs consume these reduced OM forms and co-exist with a host of chemoautotrophs. The presence of FAs in somatic tissues of filter-feeding shrimp that could only originate from microbes is strong evidence that microbial biomass is directly transferred to higher-order metazoans. This microbial loop is unique from that of the oligotrophic oceans in that it contains a methane sink, but is likely to be similar to other groundwater systems, where evidence for a similar biogeochemistry has been reported[23, 25, 58]. The generic model of ecosystem function presented here provides baseline information for future studies aiming to quantify the magnitude of this unaccounted for "upside-down" methane sink and to describe the external factors that alter the internal biogeochemistry of subterranean estuaries within karst coastal aquifers.

## Methods
**Study sites and seasons**. Between 2013 and 2016, five field campaigns were conducted to investigate flooded cave networks accessible through Cenote Bang

(the primary study site; Fig. 1; Supplementary Fig. 1) within the Ox Bel Ha Cave System and secondary locations (Supplementary Fig. 5). A comprehensive listing of the samples collected as part of this study is provided in Supplementary Data 1. Three sampling events took place during the dry season (typically December through April) and two during the rainy season (typically May through November). There was no visible evidence that any of the sites had been altered by direct or indirect human activity. Access to Cenote Bang was restricted to research team members during the study. All divers involved with the project followed protocols established by the American Academy of Underwater Sciences and the National Speleological Society Cave Diving Section.

**Physicochemical water column parameters**. Temperature, salinity, and DO were measured along vertical profiles in the water column of the caves and cenotes using a YSI XLM-600 and EXO-02 multi-parameter data sonde with a measurement frequency of 0.25–1 Hz. The sonde was carried by the lead diver, slowly descending (2–4 cm sec$^{-1}$) and advancing with the probes projecting forward to ensure an undisturbed profile of the water column.

**Sample collection and processing**. Water samples for geochemical analysis of dissolved materials were collected near the sonde profile locations in plastic 60 ml syringes fitted with 3-way stopcocks. The syringes were rinsed with distilled water and dried prior to the dive, and flushed with sample water prior to closing the stopcock. Samples for POC and lipid analyses were collected in 10 liters collapsible Nalgene carboys rinsed with distilled water prior to the dive. Samples for microbial DNA sequencing were collected in 1 liter collapsible Nalgene carboys acid washed prior to the expedition. It was not possible to rinse the carboys with sample water while underground. Because the water column in the subterranean estuary is extremely stratified (Fig. 2), data from samples collected in this study represent the water masses, not the interfaces between them where the carbon-transforming biogeochemical reactions are expected to be most active.

Samples were kept cool during transport to the field lab and processed within 8 h of collection. Samples for aqueous geochemistry were handled and stored as indicated in Supplementary Table 6. Among those, the serum vials for methane water samples were prepared prior to sample collection by adding the preservative (0.2 ml 1 M NaOH) into the empty vial, sealing the container with 1 cm thick butyl septa, and vacating the vial of air with a pump. The water sample was then transferred through the septum with a 20-gauge syringe needle. Water samples for POC, lipid, and rRNA analyses were vacuum filtered through 47 mm diameter glass fiber filters (Gelman GF/F; 0.7 µm mesh or nominal pore size), 47 mm diameter PVDF membrane filter (Millipore hydrophilic Durapore; 0.2 µm pore size), and 47 mm diameter PES membrane filter (Pall Supor membrane; 0.2 µm pore size), respectively, until the sample was exhausted or until a reduced filtration rate indicated sufficient material was collected (2–9.5 l). The filters were transported on dry ice and stored frozen at –20 °C until further analysis. Specimens of stygobitic (cave-limited) filter-feeding atyid shrimp from the genus *Typhlatya* (*T. pearsei*, *T. mitchelli*, and one specimen of *T. dzilamensis*) were collected from six locations (Supplementary Fig. 5; Supplementary Table 5). Surface dwelling insects were collected from the forest floor around the Cenote Bang. Within 6 h of collection, shrimp specimens were taxonomically identified, wrapped and stored at 0 °C in prebaked (450 °C for 4 h) aluminum foil. The specimens were transported frozen on dry ice, and then stored in the laboratory at –20 °C.

**Geochemical analysis**. Geochemical analyses were performed at the Woods Hole Oceanographic Institution (WHOI) and U.S. Geological Survey (USGS) in Woods Hole, MA, USA. Headspace methane concentrations were determined using a Shimadzu 14-A gas chromatograph (GC) equipped with a flame ionization detector. Methane was isothermally (50 °C) separated from other headspace gases with a Poraplot-Q stainless steel column (8 ft × 1/8'' OD) packed with 60/80 mesh and quantified against certified gas standards with a relative standard deviation (RSD) of 2.8% or less. Headspace concentrations were converted to dissolved concentrations using the method of Magen et al.[59] The stable carbon isotope composition of methane from the headspace of the serum vials was determined using a Thermo-Finnigan DELTA$^{Plus}$ XL isotope ratio mass spectrometer (IRMS) coupled to an Agilent 6890 Gas Chromatograph (GC) via a Finnigan GCCIII combustion interface. Variable volume (1–15 ml) gas samples, depending on concentrations, were introduced through a gas sampling valve into a 1 ml min$^{-1}$ He carrier gas stream. Methane and other condensable gases were trapped on fused silica capillary packed with 80/100 mesh Poraplot-Q immersed in liquid nitrogen. The gases were thermally desorbed from the column at 150 °C and separated on a 30 m, 0.32 mm ID Poraplot-Q column at –40 °C prior to being oxidized to $CO_2$ and analyzed by IRMS. The $^{13}C/^{12}C$ ratios of methane are expressed in the standard δ-notation using tank $CO_2$ referenced to the Vienna Pee Dee Belemnite (VPDB) standard. The standard deviation (1σ) of a 1% $CH_4$ standard analyzed at least every eight samples was 0.3‰.

For the DOC subsamples, 1:1000 trace metal grade 12 N HCl:$H_2O$ volume[60] ratio was added prior to analysis to achieve pH < 2. DOC concentration and $\delta^{13}C$ were analyzed by high-temperature combustion-isotope ratio mass spectrometry (HTC-IRMS) at the USGS-WHOI dissolved carbon isotope lab (DCIL). The DCIL HTC-IRMS system consists of an OI 1030C total carbon analyzer and a Graden

molecular sieve trap interfaced to a Thermo-Finnigan DELTA[plus] XL IRMS. Stable carbon isotope ratios are reported in the standard δ notation relative to VPDB and were corrected by mass balance to account for the analytical blank, which was less than the equivalent of 15 μM DOC in the sample. By comparison, the blank-corrected sample DOC concentrations ranged from 15.3 to 851 μM. Thus, the blank correction ranged from 6–50% of sample concentrations. DOC concentration was calculated using a standard curve consisting of four potassium hydrogen phthalate (KHP) calibration standards quantified against the mass-44 peak on the IRMS[60]. Peak areas were corrected for analytical blanks determined from ultrapure lab water injections. The concentration RSD was 5.5% during run 1 and 12.5% during run 2. The analytical error of the δ13C-DOC analysis ranged from <0.3 to 0.6‰.

DIC concentrations were determined with a Model 5011 UIC coulometer and quantified relative to a sea water certified reference material (CRM). After the addition of 100 μl 20% phosphoric acid, $CO_2$ was stripped with UHP $N_2$, delivered to the analyzer and measured with an RSD of 4% relative to the CRM value (2.2 mM). Prior to the stable carbon isotope analysis of DIC, 50 μl of 85% phosphoric acid was added to the headspace of the sample vial to allow the DIC to transfer into the headspace as $CO_2$. Samples were shaken vigorously at least once every 15 min for 2 h. Headspace gas from the sample vial was transferred and injected with a 100 μl glass syringe into a Hewlett Packard 5890 GC, where the $CO_2$ was separated isothermally (50 °C) on a Poraplot-Q capillary column (30 m, 0.32 mm ID) before isotopic analysis with the Thermo-Finnigan DELTAplus XP IRMS, as described above, with analytical error (1σ) of 1.1‰.

Sulfate and chloride concentrations were determined using a Metrohm 881 Compact Plus ion chromatograph (IC) equipped with a Metrosep A Supp 5-250 anion column. Samples from the MFW, MBW, and SGW were diluted by factors of 31, 61 and 101, respectively. Peak areas for sulfate and chloride were quantified against equivalently diluted International Association for the Physical Sciences of the Oceans (IAPSO) standard sea water analyzed at the beginning of the run and after every fifth sample. Chloride concentrations (mM) were converted to mg l⁻¹ and multiplied by 0.0018066 to determine salinity (psu). The analytical error for dissolved constituents was ±3.5% of the IAPSO standard sulfate and chloride values.

**Conservative mixing calculations.** Similar to other studies[11, 47], conservative mixing models were used to differentiate the roles of physical mixing and in situ reactions on the concentration and isotopic composition of biogeochemically reactive species through the salinity gradient of the subterranean estuary. Internal production yields an excess of the constituent relative to conservative mixing between freshwater and saline end members, while consumption results in depletion. Conservative mixing calculations for methane, DOC, DIC, and sulfate between the shallow low salinity (MFW) and deep high-salinity (SGW) ground-water layers were done using an established approach[38], adapted for the density-stratified groundwater:

$$C_{MIX} = f_{MFW}C_{MFW} + (1 - f_{MFW})C_{SGW} \quad (1)$$

Here C denotes concentration of the relevant chemical constituent and subscripts MFW and SGW represent the respective water masses used as end members, whereas the subscript MIX denotes the water mixture consisting of the two end members. $f_{MFW}$ is the fraction of the fresh end member present in the mixture calculated from the chloride concentration:

$$f_{MFW} = \frac{[Cl^-]_{SGW} - [Cl^-]_{MIX}}{[Cl^-]_{SGW} - [Cl^-]_{MFW}} \quad (2)$$

where [Cl⁻] denotes chloride concentrations, and the subscripts are the same as above. Using equations (1) and (2), we calculate the conservative mixing regimes for methane, DOC, DIC, and sulfate, adopting as end member concentrations the values from the low salinity water mass at shallow depths (in MFW) and high-salinity water mass at the deepest accessible parts of the cave (in SGW). The large chloride concentration differences between the shallow portion of the groundwater and the deeper part permits application of the method over a vertical length scale of meters in the subterranean estuary, in contrast to a horizontal length scale of kilometers typical of surface estuaries. In this study, chloride content was converted to and expressed as salinity (psu) by multiplying chloride concentrations (mg l⁻¹) by 0.0018066. Conservative mixing was calculated between the lowest $C_{MFW}$ and lowest $C_{SGW}$, as well as between the highest $C_{MFW}$ and highest $C_{SGW}$ end members for each sampling event. These calculations are represented as conservative mixing lines (CMLs) on the salinity-property diagrams. We report the absolute highest and absolute lowest results of the mixing calculations for each constituent across all seasons. The area in between the two reported mixing calculations is considered the general mixing field that incorporates all mixing lines (CML$_{area}$), where the distribution of the constituent is most likely determined by physical mixing. Positive excursion from the CML$_{area}$ shows in situ production of a chemical species, whereas negative excursion demonstrates consumption.

Stable carbon isotope mixing diagrams for methane, DOC, and DIC were used to identify the isotopic composition of constituents produced in the mixing field or isotopic fractionation associated with their removal. Conservative stable isotope

mixing models were calculated using described methods[47], also adapted for density-stratified groundwater:

$$\delta_{MIX} = \frac{f_{MFW}C_{MFW}\delta_{MFW} + (1 - f_{MFW})C_{SGW}\delta_{SGW}}{C_{MIX}} \quad (3)$$

where the subscripts are the same as in previous equations, and δ denotes δ13C values of the constituents. Similar to the conservative approach taken above, the mixing field is determined by the two extreme CMLs that were observed when generating an ensemble of the mixing lines based on solute concentration data for all samplings.

All concentration and isotope mixing diagrams were constructed with log scale on the y-axis to illustrate the full extent of the vertical salinity gradient through the three water masses (MFW, MBW and SGW). Log scale was also applied on the x-axis of methane and DOC because of the extreme differences in their concentrations across the salinity gradient.

Comparing the measured values within the mixing region to the conservative mixing models allowed us to estimate the percentage of methane and DOC removed in the shallow portion of the coastal aquifer. For this model, we assume: (1) environmental conditions (e.g., overlying vegetation, permeable bedrock matrix, passage morphology, and groundwater flow) do not change laterally in the inland portion of aquifer; (2) each sampled water layer (MFW, MBW, SGW) is representative in terms of general redox and OM conditions in that salinity regime across the inland portion of the aquifer; and therefore (3) the variation observed along the vertical salinity gradient is primarily the result of biogeochemical processes whose activity is horizontally homogenous; (4) the primary biogeochemical processes influencing the concentrations of DOC and methane are resulting in the production (OM degradation or methanogenesis) in the MFW and consumption (heterotophy or methanotrophy, respectively) in the MBW. With these assumptions, we calculated the net percent loss of reduced organic carbon due to biological oxidation of methane (methanotrophy) and DOC (heterotrophy) with respect to concentrations expected if physical mixing were the only process that modulates the distribution of $CH_4$ and DOC in the water column, using the following equation:

$$\% \text{ constituent consumed due to oxidation} = \frac{C_{MIX} - C_{MBW}}{C_{MFW}} \times 100 \quad (4)$$

where $(C_{MIX} - C_{MBW})$ determines the reduction in the concentration of the constituent due to microbial oxidation. $C_{MBW}$ and $C_{MFW}$ are the averaged measured constituent concentrations in the MBW and MFW.

**Environmental lipid biomarkers.** Respiratory quinones were extracted using a modified Bligh and Dyer extraction[61, 62] with DNP-PE-C$_{16:0}$/C$_{16:0}$-DAG (2,4-dinitrophenyl phosphoethanolamine diacylglycerol; Avanti Polar Lipids, Inc., Alabaster, AL) as internal standard and analyzed using a Thermo Q Exactive Orbitrap high-resolution mass spectrometer (Thermo Fisher Scientific, Waltham, MA, USA) equipped with an electrospray ion source (ESI) connected to an Agilent 1200 high-performance liquid chromatography (HPLC) system (Agilent, Santa Clara, CA, USA). Detection of quinones was achieved using positive ion ESI, while scanning a m/z range from 100 to 1500. The mass spectrometer was set to a resolving power of 140,000 (FWHM at m/z 200) and to 17,500 for MS2 scans. Every analysis was mass calibrated by lock mass correction. The full scan mass resolution setting corresponded to an observed resolution of 75,100 at the m/z 875.5505 of our internal standard, DNP-PE. Ion source and other full scan mass spectrometry parameters were set according to established protocols[63]. MS2 spectra were obtained in data dependent mode. For each MS full scan, five ions of highest intensity were selected in series using the quadrupole for MS2 fragmentation (4 Da isolation window) with a Stepped Normalized Collision Energy of 20, 50, and 80. Analytes were separated using reversed phase HPLC on an C$_8$ XBridge column (2.1 × 150 mm, 5 μm particle size, Waters Corp., Milford, MA, USA) as described in Collins et al.[63], modified after Hummel et al.[64] Quinones were identified by retention time, as well as accurate molecular mass of proposed sum formulas in full scan mode and tandem MS fragment spectra (Supplementary Fig. 7). Integration of peaks was performed on extracted ion chromatograms using an isolation width of 4 ppm and included the [M + H]⁺, [M + NH₄]⁺, and [M + Na]⁺ ions. Quinone abundances were corrected for the relative response of ubiquinone (UQ$_{10:10}$) standard (Sigma Aldrich, St. Louis, MO, USA) vs. the DNP-PE standard.

**Bulk stable isotopic analysis.** Prior to stable carbon isotopic analyses, particulate OM filters, soil, and invertebrate samples were exposed to 10% HCl to remove inorganic carbon, rinsed with ultrapure water, dried, and wrapped in baked (at 450 °C for 4 h) aluminum cups[30]. Fauna and soil samples were analyzed for 13C and D (2H), and POC for 13C at the University of Alaska Fairbanks (UAF) Stable isotope facility using established internal protocols δ13C values were measured by Elemental Analyzer Isotope Ratio Mass Spectrometry (EA-IRMS) using a Thermo Fisher Scientific Elemental Analyzer (Flash 2100) combined with Thermo Fisher Scientific Delta V[Plus] isotope ratio mass spectrometer and a Conflo IV interface. δ13C values are reported in reference to international isotope standards. The 44 m/z peaks were used to quantify the C content of the sample. Samples for δ2H values were analyzed on an ANCA-GSL elemental analyzer (Sercon, Crewe, UK) coupled to a Geo20–20 continuous flow IRMS at Iso-Analytical and on a Finnigan

ThermoQuest thermochemical reactor elemental analyzer (TCEA; Finnigan ThermoQuest, Bremen, Germany) attached via a Conflo III to a Thermo-Finnigan Delta$^{Plus}$ IRMS. The analytical procedures for D analysis followed previously published protocols[26, 65]. All stable isotope ratios are reported using the delta (δ) notation expressed in units per mill (‰) and D results are expressed relative to Vienna Standard Mean Ocean Water (V-SMOW). Standard deviation of δ$^{13}$C was ±0.04‰, and of δD was better than ±1.9‰. POC concentrations were quantified by comparing the response of the mass-44 peak area from the samples to peptone standards of known carbon content.

**Contribution of methane-derived carbon to the biomass.** A simple two-source mixing model[38] was used to calculate relative contributions of methane-derived carbon and the soil-derived carbon (DOC and POC) in the shrimp tissue. The following equation was used for this calculation:

$$\% \text{ methane carbon contribution in biomass} = \frac{\delta_{shrimp} - \delta_{OM}}{\delta_{methane} - \delta_{OM}} \times 100 \qquad (5)$$

where $\delta_{shrimp}$ is the measured δ$^{13}$C value of the shrimp, $\delta_{methane}$ is the average δ$^{13}$C-CH$_4$ value of methane in the MFW (−66.3 ± 0.7‰, Table 1), $\delta_{OM}$ is the average δ$^{13}$C value of DOC in the MFW (−28.0 ± 0.1‰), which is assumed to represent the isotopic content of soil-derived OM. This calculation does not consider carbon isotope fractionation by methanotrophic bacteria[66], the presumed dietary source of methane-derived carbon for the shrimp.

**Lipid biomarkers from fauna.** We performed compound-specific stable carbon isotopic analysis of membrane-bound FAs extracted from tissue of *Typhlatya* specimens. The examined tissue was removed from under the carapace and did not contain gut material. Lipid biomarkers were extracted according to a modification of established methods[67]. Double bond positions were determined through analysis of their dimethyl–disulfide adducts[68]. Two specimens were selected for this study, one with the lowest (3%) and another with highest (55%) calculated contribution of methane-derived carbon to their biomass. The δ$^{13}$C values of FA biomarkers and their percentage contributions to the total FA pool extracted from the tissue of the two shrimp specimens are listed in Supplementary Table 5. Reproducibility was monitored by repeated injections and monitoring of internal standards. Reported δ$^{13}$C values have an analytical error of ±1%.

**Phylogenetic analysis and sequence processing.** DNA was extracted from ¼ of a 47 mm diameter 0.2 μm pore size filter (Pall Supor) using a PowerViral Environmental RNA/DNA Isolation Kit (MoBio, Carlsbad, CA) following the manufacturer's recommendations. DNA was eluted into 50 μl of elution buffer and stored at −20 °C. Eluted DNA quality and quantity were evaluated on a NanoDrop ND-100 Spectrophotometer (Thermo Fischer Scientific, USA). The hypervariable V4 region of 16 S rRNA was amplified using modified 515F and 806R primers (Earth Microbiome Project; April 2015). Primers for two-step PCR amplicon barcoding library preparation were designed using the TaggiMatrix spreadsheet. Briefly, internal fusion PCR primers were constructed with the priming region for the 16S rRNA locus, a variable length tag (5–8 bp), and a 5′ sequence to target for further TruSeq library preparation. The resulting PCR1 products were purified using AMPure XP Beads (Agencourt, Beckman Coulter, USA). PCR2 was used on cleaned PCR1 products to complete TruSeq library fragment and Illumina indexing. Ampure XP cleanup was conducted, libraries were assessed for quality on a BioAnalyzer 2100, quantified on Qubit 2.0 and qPCR was conducted using the New England Biolabs Illumina Library Quantification kit. The library was sequenced on an Illumina MiSeq at the Core Facility for Nucleic Acid Analysis at the University of Alaska Fairbanks. Amplicons derived from sequencing were processed using the DADA2 R-package[69]. This package implements filtering of low-quality sequences using Q20 individual nucleotide cutoff, merging of paired-end reads, and chimera identification. Reads <150 bp were removed from the analysis and only samples with more than 3000 high-quality reads were included in down-stream analyses. Taxonomic identification was assigned also in the DADA2 package using RDP[70] as the reference database. We determined functional (metabolic) groups by using RDP to search for representative sequences from each of the operational taxonomical unit.

**Data availability.** Demultiplexed reads were deposited in NCBI Sequence Read Archive (SRA) database under accession number SRP109857. Additional data referenced in this study are tabulated in Supplementary Tables, and available through the USGS ScienceBase-Catalog at https://doi.org/10.5066/F7DJ5DJW, or on request from the corresponding author (D.B.).

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

## Acknowledgements

Funding for T.M.I. and D.B. was provided by TAMU-CONACYT (project no: 2015-049). D.B. was supported by Research-in-Residence program (NSF award #1137336, Inter-University Training in Continental-scale Ecology), Cave Research Foundation Graduate Student Grant, Cave Conservancy Foundation PhD Fellowship, Ralph W. Stone Fellowship (National Speleological Society), Grants-in-Aid of Graduate Student Research Award (Texas Sea Grant College Program), and Boost Fellowship (Texas A&M University at Galveston). Additional financial support was provided by NSF DEB-1257424 (M.B.L. and M.C.L.), the Postdoctoral Program at Woods Hole Oceanographic Institution and U.S. Geological Survey (K.W.B.). We thank José Luis Villalobos, Sergio Benitez, Olinka Cortes, Brett Gonzalez, Jacob Pohlman, Jake Emmert, and István Brankovits for assistance with field expeditions, and Moody Gardens (Galveston, Texas) for supporting the field work. We also thank Pete van Hengstum for help with art work, and the late John Hayes for productive discussions and guidance during the development and preparation of the study and the manuscript, and Benjamin A.S. Van Mooy for access to the HPLC/ESI-MS. Sean P. Sylva, Michael Casso, and Ian C. Herriott helped with laboratory analyses. Any use of trade names is for descriptive purposes and does not imply endorsement by the U.S. government.

## Author contributions

D.B. and J.W.P. designed the study and prepared the manuscript; D.B., J.W.P., T.M.I., and B.P. collected the samples; D.B., J.W.P., H.N., M.B.L., M.C.L., M.F.L., and K.W.B. performed the experiments and data analysis; F.A. contributed samples and data; all authors contributed to the editing of this work.

## Additional information

**Competing interests:** The authors declare no competing financial interests.

