## [Peer Review File · Nature Communications]

Reviewers' comments:

Reviewer #1 (Remarks to the Author):

The manuscript builds on the previous work of the authors (Pohlman et al 1997) to propose a previously unforeseen microbial loop, based on methane and dissolved organic matter (DOM), exists within the anchialine karst estuary of the Yucatan Peninsula. Central to this work is the suggestion that DOM generated within the jungle soils of the Yucatan enters the meteoric lens of the aquifer (supported by the dramatic rise in DOM during rain events), where it fuels microbial growth and through subsequent consumption, organisms in higher trophic levels. This is not really a new concept in of itself, as it is well established that surface organic material (whether naturally-derived DOM or pollutants) can enter groundwater and influence microbial growth. Rather, the proposed novelty comes from the arguments that a significant amount of the organic carbon driving the microbial loop is derived from methane.

I find this latter claim difficult to justify – firstly due to the relatively oxidized nature of methane compared to the other forms of DOM from the breakdown of plant matter that would be available for microbial use (methane has an E_o' -0.25 V for 6 e⁻ compared to glucose E_o' -0.43 V for 24 e⁻). Much of the case for methane comes from the isotopic fractionation of the carbon observed in the shrimp and the observed microbiology. I found that interpretation of microbiology was limited. For example, the presented phylogeny grouped >90% of the identified phylotypes into 'heterotrophs'. What does that even mean? Most bacteria encode the capacity to carry out a number of different metabolic strategies, depending on the energy sources available. Even within genera the ability to carry out heterotrophic versus chemolithotrophic capabilities may not be fully defined. A detailed description of the phyla found, as well as a detailed justification as to why they were placed in use metabolic 'groups' should be include.

Beyond this, I was somewhat concerned on the the over-reliance on the idea that heterotrophy only occurs through respiratory oxidation. Given that there was a significant amount of DOM in anoxic conditions with limited electron acceptors available, I suspect that fermentation is playing an important role in carbon turnover. There is also no mention of the potential of nitrate to serve as an electron acceptor (as described in Pohlman et al., 1997) or the potential use of methanol by methanotrophs. Undoubtedly methanol is being produced in the breakdown of the organic matter in the jungle above and might actually be a higher potential source of energy for the observed (but undescribed) methanotrophs, given the very low numbers of methanogenic species observed.

I also had one question regarding sulfate reduction, which the authors alluded to in the paper, but didn't elaborate on. There are two types of cenote in the Yucatan – those that are part of a flowing, anchialine estuary, and the more isolated meromictic cenotes. In the case of the meromictic cenotes, which are still supplied with seawater through the aquifer, the stratification leads to the development of both a significant sulfate-reduction zone, as well as the production of a clear H₂S oxidation layer just above the halocline. Indeed, the microbes in the H₂S layer produce so much elemental sulfur that a milky 'cloud' appears just above the halocline and is a significant feature noted by the divers, even in some of the anchialine caves. The isotopic fractionation of this S₂⁻ is dramatic (63.2% - Socki et al, 2002); however, I have no idea how this affects the carbon fractionation.

Also within the anchialine caves, tidal pumping pushes seawater inland. Is it not possible that this tidal pumping could obscure the sulfate-reduction signature that the authors look for in supplemental figure 3? Indeed, the observation sulfate-reducing microbial species in their analyses would rather suggest that such activities could be driving organic breakdown in these caves, and even potentially contributing to the lipid signatures seen the in the shrimp (In 343-344).

In all, I like the idea of an 'upside-down methane sink' that has not previously been described, but I am not sure that the authors have really made the case for its existence or what fraction of the

total carbon pool this accounts for. Given the data, I'm more inclined to believe that it is a relatively small contribution, which may be further undermined by the role of methanol. As such, the novelty of the described system remains unclear.

Reviewer #2 (Remarks to the Author):

Overall this manuscript presented a very interesting and novel description of the carbon flow in subsurface cave systems. They found that methane and DOC from within the system were dominant energy sources for the food web and that was reflected in the microbial community, the animal food web, as well as the biogeochemistry of the system. The authors did a very comprehensive treatment of the system applying many different techniques to present a holistic view, which they then advance into a model of an inverted water system. These include some modeling approaches which show that these systems are likely sinks to methane that are currently under-appreciated. I have only minor comments on the MS although there was a particular error that should be addressed concerning their presentation of fatty acid synthetic pathways (which does not impact the validity of the conclusions).

There were many great bits in here. Showing that that acetoclastic methanogenesis was the dominant form of methanogenesis. Modeling to estimate the rate of turnover of some of these C-pools. Just names a few.

L 338 – this is not correct. 16:1w7 and 18:1w9 are both readily synthesized by animals/metazoans. 16:1w5 is not nor are the odd chain FAs listed. None of the references cited are truly appropriate for this statement anyway. Suggest Dalsgaard et al. *Advances in Marine Biology* 46 2003 for an overview. As follows, 339 "suggesting...symbiosis" is not correct. The FAs that are referenced have been found in SRB bacteria but are also found in many bacteria. This incorrect statement does not impact the findings of the paper and should be omitted or corrected.

Overall, all of the sections are well written and flow nicely into each other. The introduction is concise and well formed. The results present their data nicely and the discussion puts in in context. In general the figures are also nice and complimentary.

Figure 5: A nice figure, however I find the $\delta^{13}C$ of the FAs and the concentrations of the FAs going in opposite directions confusing. I would invert the top figure of B to be consistent. In A, the key (i.e. the picture of the shrimp) is confusing. Are those data points in the key just the key or are they data points? Also, in B it says there is a shrimp with a bulk $\delta^{13}C$ value of $\sim 48-50$ per mil, is that in figure A? It doesn't appear so.

Table 1: This should report Standard Error not Standard Dev unless they are instrumental replicates rather than replicate samples of the environment.

Reviewer #3 (Remarks to the Author):

I am familiar with the techniques being presented but am not a specialist in most of them and so my comments pertain to the methodological approach taken by the authors to elaborate the workings of this complex system in a challenging environment.

This paper will be of key interest to researchers working on the ecology and geochemistry of anchialine systems. It will also attract attention from several disciplines and to managers dealing with many aspects of the coastal karst systems that comprise a quarter of the world's coastline, as well as those interested in submarine groundwater discharge.

Coastal karst and their contained aquifers and subterranean estuaries (often referred to as

anchialine systems) are of great importance as a water resource but intrinsically vulnerable to anthropogenic insults including rising sea level. Anchialine systems are known for very diverse crustacean assemblages with highly predictable generic composition of biogeographic and/or phylogenetic relicts distributed across the tethyan realm. These characteristics make anchialine systems the focus of considerable research effort in phylogenetics and historical biogeography but the attributes also attracts research amongst a broad range of disciplines, particularly, concerning the energy system that allows the coexistence of this suite of specialised species in an oligotrophic habitat.

Brankovits et al. have focused on elucidating the sources and pathways of trophic energy in an anchialine cave in Yucatan which they do so comprehensively and with elegance. The research is methodologically robust and is presented with exceptional clarity and is lucidly presented with comprehensive data using prolific lines of evidence which are also well supported and clearly presented in the supplementary data.

The system they have selected is remarkably stable, a consequence of the anoxic zone being the uppermost, lacking the large vertical migrations of anoxic zones found in some anchialine and lacustrine systems.

Lentic ecologists are often askance at the inverted DO profiles found in deep anchialine systems, one maintained by allochthonous DO in the marine derived water. The authors identify a previously unrecognised methane sink in a karst subterranean estuary for methane formed by acetoclastic methanogenesis with methane oxidising bacteria being the major food source for the anchialine fauna with carbon from methane able to comprising the bulk of their diet. This is the first demonstration of such a system and one that is inverted relative to similar systems in open water. The research refocuses understanding of both ecosystem function and the carbon cycle in these systems. But they go much further, providing a general model for such systems that is consistent with the sparser data known elsewhere.

The use of end-point mixing models throughout provides a robust analysis of amounts expected at given levels but is not overplayed and where the expected amounts could lack sufficient resolution for the substance this is clearly stated. Such clarity pertaining to the confidence in the results is seen throughout the paper.

One might discuss potential implications/similarities with other systems, such as ice-covered groundwater flow-through lakes, or heterogeneous aquifers where GW flow rate increases with depth.

This paper is tightly written and well documented. I have few specific comments.

Comments:

114 -- DO in the POOL is not always elevated relative to the MFW and MBW (see Fig. 2B 2014 Aug).

227-229 -- While the hypothesis is entirely consistent with the data and with the general understanding of carbonate karst, neither the point of methanogenesis nor the permeability of the limestone is subsequently demonstrated; a method could be proposed to test this.

334-338 -- This argument is not clearly demonstrated in Fig 5B and needs to be clarified. The relative abundance differences are not heavily weighted to one specimen (ca 4:3).

Reviewers' comments in plain text and *author's responses italicized*:

We would like to express our gratitude for the helpful and constructive comments of the three anonymous reviewers. To address their remarks, we responded below and modified the manuscript accordingly (please find attached the revised version).

Reviewer #1 (Remarks to the Author):

The manuscript builds on the previous work of the authors (Pohlman et al 1997) to propose a previously unforeseen microbial loop, based on methane and dissolved organic matter (DOM), exists within the anchialine karst estuary of the Yucatan Peninsula. Central to this work is the suggestion that DOM generated within the jungle soils of the Yucatan enters the meteoric lens of the aquifer (supported by the dramatic rise in DOM during rain events), where it fuels microbial growth and through subsequent consumption, organisms in higher trophic levels. This is not really a new concept in of itself, as it is well established that surface organic material (whether naturally-derived DOM or pollutants) can enter groundwater and influence microbial growth. Rather, the proposed novelty comes from the arguments that a significant amount of the organic carbon driving the microbial loop is derived from methane. I find this latter claim difficult to justify – firstly due to the relatively oxidized nature of methane compared to the other forms of DOM from the breakdown of plant matter that would be available for microbial use (methane has an $E_o' -0.25$ V for 6 e- compared to glucose $E_o' -0.43$ V for 24 e-).

RESPONSE: *We agree with the reviewer that other studies have shown that surface organic matter influences microbial growth and the trophic status of the higher-order food web in groundwater ecosystems. Such studies have primarily focused on inland systems and pollutants with significantly less attention on coastal aquifer and subterranean estuary ecosystems and have been conducted with less analytical rigor than this study. As a result, Pohlman et al.'s (1997) work has remained the state-of-the-art study in pristine karst estuaries, leaving us ignorant with respect to the origins and composition of organic matter (OM) that fuels the anchialine fauna, a community of invertebrates found worldwide in subterranean estuaries along coastlines.*

*The evidence that methane is a significant source of carbon and energy for the food web is based on diverse geochemical, microbiological and lipid biomarker datasets. Among those data, the most compelling justification that methane contributes substantially to the higher-order food web is the ^{13}C and D-depleted nature of the *Typhlatya* shrimp biomass. There are no other known carbon sources capable of producing that combination of ^{13}C and D-depleted values in higher-order consumers. Furthermore, we carefully trace the ^{13}C -depleted carbon source from its origins to the terminal consumers. Specifically, we observe a ^{13}C -depleted methane source in the anoxic*

freshwater portion of the passages that becomes more enriched with ^{13}C in the hypoxic portion of the passages (evidence of oxidation). We trace that unique methane-source isotopic signal into fatty acids and bulk biomass of the shrimp. Furthermore, the most ^{13}C -depleted fatty acids are known to be associated with methane oxidizing bacteria. With respect to the D-depletion, we only measure this in the higher-order consumers, but it remains uniquely consistent with a methane source (e.g., Grey, 2016). This approach has been proven as a robust and reliable strategy for linking methane from deep-ocean seeps to deep-ocean fauna (Niemann et al., 2013). There are no other known sources of organic matter in this system that could explain our multiple observations. We are therefore highly confident that a significant amount of the carbon entering higher levels of the food web is derived from methane and that microbes play a role in the energy transfer. We are confident about this interpretation even if it is evident that other OM sources are also present in the system.

We share the view with the reviewer that the DOC pool consists of a wide range of compounds, some containing more energy than methane (e.g., glucose). However, freshwater DOM tends to be dominated by humic compounds that are not easily incorporated into the food web. Although the redox potential of methane is less than other DOM components, multiple studies have shown ecosystems are adapted to its consumption, and that aerobic oxidation is an efficient mechanism for food web carbon assimilation (e.g., Kankaala et al., 2006). That said, we also agree future studies are needed to characterize and quantify the role of other DOM constituents supporting the anchialine food web via the microbial loop. Methanol – its cycling closely linked to methane oxidation – is likely to be one of them.

Much of the case for methane comes from the isotopic fractionation of the carbon observed in the shrimp and the observed microbiology. I found that interpretation of microbiology was limited. For example, the presented phylogeny grouped >90% of the identified phylotypes into 'heterotrophs'. What does that even mean? Most bacteria encode the capacity to carry out a number of different metabolic strategies, depending on the energy sources available. Even within genera the ability to carry out heterotrophic versus chemolithotrophic capabilities may not be fully defined. A detailed description of the phyla found, as well as a detailed justification as to why they were placed in use metabolic 'groups' should be include.

RESPONSE: *As its pointed it out in the review, the isotopic fractionation of the carbon observed in the shrimp and the observed microbial assemblages were indeed interpreted as patterns consistent with the presence of methane, however, our argument for active methane oxidation and methane uptake by the food web stands on direct geochemical evidence in addition to molecular genetics.*

First and foremost, the concentration and isotopic profiles of CH_4 and the conservative mixing model (Fig 3. A-B) provide strong and clear evidence for an active methane (and DOC, Fig. 3 C-D) removal mechanism in the water column (In. 261-265). It is well established that the observed concentration and isotopic trends can only be mediated via methanotroph microbial activity (Whiticar 1999; In. 268-270). Considering the inverted structure of the stratified groundwater layers (In. 230-234), this description based on aqueous geochemistry (in itself) provides an argument for the presence of an "upside-down" methane sink mechanism. For all these reasons, we argue that the mixing calculations provide an indirect (but important and robust) evidence for the presence of microbes capable of methanotrophy.

*As a next step, we intended to unequivocally corroborate the existence of microbial assemblages that are potentially capable of mediating the above described biogeochemical processes by providing direct evidence for their presence with molecular and lipid biomarker analyses. We completely agree that the microbial community has a widespread metabolic capacity (see next response section below for how we clarified our statements about heterotrophy). We appreciate that the reviewer has drawn our attention to the limited interpretation of the microbial taxonomic diversity. Following Reviewer #1's comments, we added a figure (**Supplementary Fig. 4**) to provide an overview of the phyla found. Moreover, we also added a table (**Supplementary Table 5**) to provide details about the phylogenetic affiliations of microbial groups found and their relative abundance.*

*Grouping microbes into operational ecological units becomes more and more common (e.g., Preheim et al. 2016 Nature Microbiol.; Schouten et al. 2008 Applied and Environmental Microbiol.). The revised text now includes: "Phylogenetic affiliations based on sequence reads from 16S rRNA genes were clustered into metabolic groups ... These functional groups were determined to identify microbes capable of mediating biogeochemical pathways inferred from geochemical analyses (**Fig. 4 A**)" (In. 159-163). We also updated the methods section for clarification, to read: "Taxonomic identification was assigned also in the DADA2 package using RDP⁷⁰ as the reference database. We determined functional (metabolic) groups by using RDP to search for representative sequences from each of the operational taxonomical unit" (In. 538-539). Finally, we clarified the definition of the most general functional groups (In. 815-819): 'Multiple' indicates there was no specific function associated with the closest match type strain. The majority of sequences were from bacteria and archaea capable of utilizing a variety of complex organic compounds ('Others'; see **Supplementary Fig. 4** for further details). For a comprehensive phylogenetic listing and relative abundance of sequence reads see **Supplementary Table 5**."*

*Finally, as the reviewer pointed out, we investigated the conspicuous $\delta^{13}\text{C}$ (and δD) signatures from the shrimp's bulk tissue (**Fig 5. A**) and $\delta^{13}\text{C}$ values of the fatty acids extracted from the shrimp's tissue (**Fig 5. B**). This approach allowed us to confirm that methane carbon present in the biomass of the shrimp is derived from microbial sources that previously utilized methane carbon (isotopic data) (In. 337-364). This observation provides another indirect (but strong) evidence for the presence of methane consuming microbes in the vicinity of the shrimp. Based on these data we conclude that microbes (i.e., members of the microbial loop) convey methane carbon to the biomass of the cave shrimp.*

Beyond this, I was somewhat concerned on the the over-reliance on the idea that heterotrophy only occurs through respiratory oxidation. Given that there was a significant amount of DOM in anoxic conditions with limited electron acceptors available, I suspect that fermentation is playing an important role in carbon turnover. There is also no mention of the potential of nitrate to serve as an electron acceptor (as described in Pohlman et al., 1997) or the potential use of methanol by methanotrophs. Undoubtedly methanol is being produced in the breakdown of the organic matter in the jungle above and might actually be a higher potential source of energy for the observed (but undescribed) methanotrophs, given the very low numbers of methanogenic species observed.

RESPONSE: *Results of the quinone biomarker analysis strongly support that organisms in the hypoxic portions of the groundwater environment (see DO profiles on **Fig. 2**) were primarily active as aerobic heterotrophs (In. 288--292; **Fig. 4 B**). Nevertheless, we certainly agree with the reviewer that*

fermentation drives organic matter decay and the generation of dissolved reduced carbon species, including methane in the shallow, anoxic portion of the aquifer. To clarify that we share this view, we modified lines 218-220 (it now reads: “Anaerobic decomposition of this soil OM from the overlying tropical jungle is the most likely source of the high concentrations of DOC ($665 \pm 132 \mu\text{M}$) and methane ($6,466 \pm 659 \text{ nM}$) measured in the cave-accessible MFW”) and 222-224 (it now reads: “...suggests that methane and DOC originate from a similar source; one that likely involves syntrophic interactions between fermentative bacteria and methanogenic archaea”). Furthermore, we suggest that acetoclastic methanogenesis (a final step of a fermentative process) is the primary source of methane in the system (ln. 336). In Fig. 6 and its caption, we conclude that the anoxic saturated zone of the carbonate rock-matrix is the primary source of DOC and methane. Fermentation of organic matter to DOM end-products other than methane (e.g. methanol) may be important in the cave waters, and is a worthy topic for future investigations. However, we are not aware of any studies showing that methanol or other fermentative products (including acetate) are sufficiently depleted in ^{13}C to account for the stable carbon isotopic content we observed in the higher order crustaceans.

We agree that denitrification as suggested by the reviewer and nitrification as suggested by Pohlman et al. (1997) are potential sources for ^{13}C -depleted biomass. However, as we explain in the revised text (lines 315-319) the carbon assimilation efficiency of these nitrogen based pathways is extremely low, and the concentrations of nitrate reported by Pohlman et al. (1997) are substantially lower than what is found in the deep ocean, where these processes do not prominently affect the deep ocean carbon cycle. We acknowledge that we are dealing with a different ecosystem that is even more carbon limited than the deep ocean, but given the relatively high concentrations of methane (relative to other aquatic and marine systems) and evidence for its oxidation (Fig. 3), the likelihood that methane is the source of ^{13}C -depletion in the higher order consumers is much greater.

I also had one question regarding sulfate reduction, which the authors alluded to in the paper, but didn't elaborate on. There are two types of cenote in the Yucatan – those that are part of a flowing, anchialine estuary, and the more isolated meromictic cenotes. In the case of the meromictic cenotes, which are still supplied with seawater through the aquifer, the stratification leads to the development of both a significant sulfate-reduction zone, as well as the production of a clear H₂S oxidation layer just above the halocline. Indeed, the microbes in the H₂S layer produce so much elemental sulfur that a milky 'cloud' appears just above the halocline and is a significant feature noted by the divers, even in some of the anchialine caves. The isotopic fractionation of this S₂₋ is dramatic (63.2% - Socki et al, 2002); however, I have no idea how this affects the carbon fractionation.

RESPONSE: *Sulfate-reduction (SR) is certainly an important respiratory mechanism in aquatic and marine systems, including some of the Yucatan Peninsula's cenotes (meromictic and/or flowing anchialine estuary-type) where anoxic conditions are dominant due to high OM inputs and/or low dissolved oxygen concentrations. In the Yucatan, these conditions most often occur near openings where OM can directly enter the groundwater (Pohlman 2011). As the reviewer pointed it out, one of the signs of dominant SR is a clearly visible H₂S cloud just above the halocline, where sinking organic matter accumulates and high concentrations of sulfate are available from the saline groundwater. In the studied cave system, the availability of organic matter (with the exception of high DOC in the fresh water mass) is very low. As stated in Table 1, the concentration of particulate organic carbon in the groundwater (cave environment) is less than 11 μM . On average (lines 147-148) POC*

concentrations are 1.6% of the DOC concentration (In. 147). Therefore, within the water column of the cave environment (isolated from particulate inputs), sinking organic matter does not appear to accumulate at the sediment water interface or along density gradients in the water column at the levels required to support the bacterial SR. Furthermore, where the POC and DOC concentrations are highest (in the meteoric fresh water, or MFW), sulfate concentrations are low (<1 mM), which further limits the thermodynamic potential of SR.

Beyond our measured geochemical observations, we found no evidence for the presence of H₂S during any of the five sampling campaigns conducted between 2013 and 2016 (In. 307). (Even low levels of H₂S are very easy to detect from its noxious smell). This basic observation was consistent with the presence of dissolved oxygen (hypoxic conditions) in the brackish portion of the meteoric lens and the saline groundwater (Fig. 2; In. 279). Because sulfate is a less favorable electron acceptor than oxygen, aerobic organisms typically outcompete anaerobes, which makes SR in the water column of the studied caves rather unlikely. **Supplementary Fig. 3 A** demonstrates that we were unable to detect either sulfate-reduction or sulfide-oxidation. From that we conclude the sulfur cycle was inactive at our study site, notwithstanding the observation that sulfate-reducing and sulfur-oxidizing bacterial 16S rRNA sequences are present in the cenote pool, fresh groundwater and brackish groundwater (see Fig. 4 A). The presence of a genetic sequence does not require that it is active. Further investigation is required to determine if these functional groups are active.

Regarding the reviewer's inquiry about how SR affects carbon isotope fractionation, carbon isotope fractionation during sulfate reduction is generally on the order of 5-10‰ in the negative direction relative to the CO₂ source (e.g., Londry and Des Marais, 2003). For example, the biomass for an SRB using the DIC with a δ¹³C of -16.4‰ from the meteoric fresh water (MFW in Table 1) would be ~ -21 to -26‰. These values are considerably more enriched with ¹³C than any of the fatty acids from the more ¹³C-depleted shrimp illustrated in Fig. 5B, further suggesting SR was not an ecologically relevant process.

We are anecdotally aware of instances in the Yucatan where SR may be active in the groundwater of the cave environment somewhat removed from the open-to-air high OM-input sinkholes, and we intend to investigate those sites in the future. However, the site we studied here was specifically selected as a representative of an organic matter limited system (Pohlman 2011) to characterize the baseline ecosystem function for a pristine system, while they still exist. These OM-limited systems are likely to be more prevalent in the vast cave networks of the Yucatan (lines 72-77) even though they are less known perhaps due to more difficult access than the open (high-OM input) sinkholes.

Also within the anchialine caves, tidal pumping pushes seawater inland. Is it not possible that this tidal pumping could obscure the sulfate-reduction signature that the authors look for in supplemental figure 3? Indeed, the observation sulfate-reducing microbial species in their analyses would rather suggest that such activities could be driving organic breakdown in these caves, and even potentially contributing to the lipid signatures seen in the shrimp (In 343-344).

RESPONSE: Indeed, tidal pumping is one of the important mechanisms transporting sulfate-rich marine-derived water into the aquifer. Since this saline groundwater is the primary source of sulfate in the system, we used it as our high salinity endmember in the mixing model (In. 466-472, **Supplementary Text 1, Supplementary Fig. 3**). We absolutely agree with the reviewer that large concentrations of sulfate derived from the saline groundwater will potentially obscure the detection

*of sulfate-reduction with the proposed method. For that reason, we state in the main text that possibly “the conservative mixing model approach we used lacks the sensitivity required to detect changes in sulfate concentration” (ln. 311-312), therefore, **Supplementary Fig. 3** should be interpreted with caution. However, as described above we found enough lines of evidence suggesting SR was not active (no evidence of sulfate consumption, no detectable sulfide and $\delta^{13}\text{C}$ values of the fatty acids more ^{13}C -depleted than expected for SRBs) or that conditions were not suitable for active SR (limited POC to accumulate at interfaces, presence of dissolved oxygen and limited [$<1\text{ mM}$] sulfate where OM was most abundant) that we presently assume the 16s rRNA sequences detected were inactive. Given these evidences, we are reluctant to conclude that sulfate-reduction is a dominant process in the studied system. We updated the text (**lines 302-323**) to clarify our reasoning and better reflect the point of view described above.*

In all, I like the idea of an ‘upside-down methane sink’ that has not previously been described, but I am not sure that the authors have really made the case for its existence or what fraction of the total carbon pool this accounts for. Given the data, I’m more inclined to believe that it is a relatively small contribution, which may be further undermined by the role of methanol. As such, the novelty of the described system remains unclear.

RESPONSE: *We are pleased to hear that the reviewer likes the ‘upside-down methane sink’ concept. As we explain above, we make the case for its existence by describing the biogeochemical pathways that produce and consume methane and DOM in this groundwater system. To our knowledge, no other study has shown the presence of similar processes in subterranean karst estuaries. Especially, we found no mention in the literature that such processes may be present in density stratified aquifers where the position of OM-rich and OM-poor water masses are reversed relative to well-known open-to-air limnic settings. **Table 1** provides a comprehensive listing of the primary organic and inorganic aqueous carbon pools. We agree that methane is a small component of the total organic carbon pool (~1% percent in the MFW). However, just like in other environments, the relative abundances of the other carbon pools do not affect the methane sink concept. The magnitude of the methane sink is independent of the magnitude of other carbon pools. A more important comparison in evaluating the concept is that the dissolved methane concentrations of $> 6,000\text{ nM}$ in the MFW are high in comparison to other aquatic systems including lakes (Holgerson and Raymond, 2016), and the ocean (Thornton et al., 2016). Therefore, these waters have the potential to contribute substantial methane to the atmosphere. Microbial methane oxidation in aerated soils (Conrad, 1995) and caves (Fernandes-Cortes et al., 2015) is an important global sink for methane. The role that aquatic methane oxidation plays in the consumption of subterranean methane in this karstic, tropical system is a novel concept we aim to introduce in this study with the hope that future studies will quantify its contribution.*

The reviewer’s suggestion that methanol plays a more important role in the carbon cycle than we have indicated is insightful. However, the role of methanol in the carbon cycle is not related to the issues of this groundwater being an unrecognized methane sink.

Reviewer #2 (Remarks to the Author):

Overall this manuscript presented a very interesting and novel description of the carbon flow in subsurface cave systems. They found that methane and DOC from within the system were dominant energy sources for the food web and that was reflected in the microbial community, the animal food web, as well as the biogeochemistry of the system. The authors did a very comprehensive treatment of the system applying many different techniques to present a holistic view, which they then advance into a model of an inverted water system. These include some modeling approaches which show that these systems are likely sinks to methane that are currently under-appreciated. I have only minor comments on the MS although there was a particular error that should be addressed concerning their presentation of fatty acid synthetic pathways (which does not impact the validity of the conclusions). There were many great bits in here. Showing that that acetoclastic methanogenesis was the dominant form of methanogenesis. Modeling to estimate the rate of turnover of some of these C- pools. Just names a few.

RESPONSE: *We are pleased to hear the reviewer's compliments about the methodological approach we took and that we have satisfied the reviewer regarding our final conclusions based on the available dataset. We appreciate the comments about our presentation of fatty acid synthetic pathways. As listed below, we addressed the reviewer's concerns and we absolutely agree that these corrections will improve the quality of the final product.*

L 338 – this is not correct. 16:1w7 and 18:1w9 are both readily synthesized by animals/metazoans. 16:1w5 is not nor are the odd chain FAs listed. None of the references cited are truly appropriate for this statement anyway. Suggest Dalsgaard et al. *Advances in Marine Biology* 46 2003 for an overview. As follows, 339 “suggesting...symbiosis” is not correct. The FAs that are referenced have been found in SRB bacteria but are also found in many bacteria. This incorrect statement does not impact the findings of the paper and should be omitted or corrected.

RESPONSE: *Following the reviewer's comments, we made substantial changes in the paragraph where we interpret the FA dataset (ln. 337 - 364). To exclude any misunderstanding that 16:1w7 and 18:1w9 FAs are exclusive to certain bacteria and to be more specific about our interpretation, we modified the text and the cited literature to read “Stable isotope probing experiments using arctic lake sediments have shown that several compounds present in the shrimp tissue (C14:0, C16:1w7 and C18:1w7) are synthesized from methane carbon by methanotrophs that were also present in the cave waters (e.g., Methylococcaceae and Methylocystacea) (He et al. 2015). These FAs and the methylated FAs are also found in other organisms, but are not likely to have the 13C-depleted isotope signatures observed here.”.*

Overall, all of the sections are well written and flow nicely into each other. The introduction is concise and well formed. The results present their data nicely and the discussion puts in in context. In general the figures are also nice and complimentary.

RESPONSE: *We are pleased to have satisfied the reviewer's expectations.*

Figure 5: A nice figure, however I find the $\delta^{13}C$ of the FAs and the concentrations of the FAs going in opposite directions confusing. I would invert the top figure of B to be consistent. In A, the key (i.e. the picture of the shrimp) is confusing. Are those data points in the key just the key or are they data points?

Also, in B it says there is a shrimp with a bulk $\delta^{13}\text{C}$ value of $\sim 48\text{-}50$ per mil, is that in figure A? It doesn't appear so.

RESPONSE: *We have made changes to avoid confusion by synthesizing the data from the two shrimp into one figure (see updated **Fig. 5 B**). This way relative abundance of FA compounds and isotopic data obtained from the two shrimp are easier to compare. To exclude any confusion, we removed the picture of the shrimp from **Fig. 5 A** (the picture is shown on **Supplementary Fig. 1**). **Fig. 5 A** does not include the specimens on **Fig. 5 B** because no deuterium data are available for those specimens (**Supplementary Table 6.**).*

Table 1: This should report Standard Error not Standard Dev unless they are instrumental replicates rather than replicate samples of the environment.

RESPONSE: *Thank you for the comment, **Table 1** has been updated to standard errors instead of standard deviations.*

Reviewer #3 (Remarks to the Author):

I am familiar with the techniques being presented but am not a specialist in most of them and so my comments pertain to the methodological approach taken by the authors to elaborate the workings of this complex system in a challenging environment.

This paper will be of key interest to researchers working on the ecology and geochemistry of anchialine systems. It will also attract attention from several disciplines and to managers dealing with many aspects of the coastal karst systems that comprise a quarter of the world's coastline, as well as those interested in submarine groundwater discharge.

Coastal karst and their contained aquifers and subterranean estuaries (often referred to as anchialine systems) are of great importance as a water resource but intrinsically vulnerable to anthropogenic insults including rising sea level. Anchialine systems are known for very diverse crustacean assemblages with highly predictable generic composition of biogeographic and/or phylogenetic relicts distributed across the tethyan realm. These characteristics make anchialine systems the focus of considerable research effort in phylogenetics and historical biogeography but the attributes also attracts research amongst a broad range of disciplines, particularly, concerning the energy system that allows the coexistence of this suite of specialised species in an oligotrophic habitat.

Brankovits et al. have focused on elucidating the sources and pathways of trophic energy in an anchialine cave in Yucatan which they do so comprehensively and with elegance. The research is methodologically robust and is presented with exceptional clarity and is lucidly presented with comprehensive data using prolific lines of evidence which are also well supported and clearly presented in the supplementary data.

The system they have selected is remarkably stable, a consequence of the anoxic zone being the uppermost, lacking the large vertical migrations of anoxic zones found in some anchialine and lacustrine systems.

Lentic ecologists are often askance at the inverted DO profiles found in deep anchialine systems, one maintained by allochthonous DO in the marine derived water. The authors identify a previously unrecognised methane sink in a karst subterranean estuary for methane formed by acetoclastic methanogenesis with methane oxidising bacteria being the major food source for the anchialine fauna with carbon from methane able to comprising the bulk of their diet. This is the first demonstration of such a system and one that is inverted relative to similar systems in open water. The research refocuses understanding of both ecosystem function and the carbon cycle in these systems. But they go much further, providing a general model for such systems that is consistent with the sparser data known elsewhere.

The use of end-point mixing models throughout provides a robust analysis of amounts expected at given levels but is not overplayed and where the expected amounts could lack sufficient resolution for the substance this is clearly stated. Such clarity pertaining to the confidence in the results is seen throughout the paper.

One might discuss potential implications/similarities with other systems, such as ice-covered groundwater flow-through lakes, or heterogeneous aquifers where GW flow rate increases with depth.

***RESPONSE:** We greatly appreciate the reviewer's rousing endorsement. Discussing the implications of the described processes relative to other groundwater system and lakes would be worthwhile addition to the current version of the manuscript. However, we are limited in our capacity to extend the discussion based on manuscript length limitations.*

This paper is tightly written and well documented. I have few specific comments.

Comments:

114 -- DO in the POOL is not always elevated relative to the MFW and MBW (see Fig. 2B 2014 Aug).

***RESPONSE:** We appreciate the reviewer's comment. After reexamining the text, however, we think that the original statement is consistent with the data presented in **Fig. 2 B**. The data from the field campaign mentioned by the reviewer as outlier (2014 Aug), just like all other seasons, show slightly elevated DO concentrations relative to the MFW and MBW. Our first field campaign (2013 Dec) was the only one when this pattern was not obvious. However, at that time DO in the POOL was generally higher than in the meteoric water within the cave environment. To clarify, we have modified the text to read: "DO in the POOL was consistently low (10 to 37 μ M), but always elevated relative to the MFW and MBW during each event" (**ln. 116**).*

227-229 -- While the hypothesis is entirely consistent with the data and with the general understanding of carbonate karst, neither the point of methanogenesis nor the permeability of the limestone is subsequently demonstrated; a method could be proposed to test this.

***RESPONSE:** To clarify, we made changes in the text and supported our statements with appropriate references needed. It now reads: "This further supports the hypothesis that the saturated zone within the permeable rock matrix (Ford and Williams 2013, Perry et al. 2002), a portion of the aquifer that was inaccessible to divers, was the most likely source of DOC in the MFW" (**ln. 227-229**).*

334-338 -- This argument is not clearly demonstrated in Fig 5B and needs to be clarified. The relative abundance differences are not heavily weighted to one specimen (ca 4:3).

RESPONSE: Thank you for drawing attention to the unclear statement. The argument regarding the relative abundance differences has been removed and the sentence was modified for clarity to read: "Even-chained saturated FAs (C14:0, C16:0, and C18:0) had compound-specific $\delta^{13}\text{C}$ values closely matching the bulk tissue (**Fig. 5 B**), suggesting the shrimp synthesized these compounds de novo from their dietary carbon sources" (**ln. 344-347**).

Reviewers' comments:

Reviewer #1 (Remarks to the Author):

I appreciate the authors taking the time to answer my myriad questions. However, even with the additional clarification and information provided by the author, my underlying concerns regarding the microbiology remains: the observed microbiology does not support the isotopic data that microbial productivity from methane oxidation supports 55% of the diet of the shrimp. I'm not arguing that the isotopic data is wrong, but there appears to be a critical missing piece of the biogeochemical cycle that leads to the observed isotopic values – unless there is some characteristic grazing behavior from the shrimp that allows them to sub-select 1% of the microbial population?

The authors argue that their isotopic mixing data provides “indirect (but important and robust) evidence of microbes capable of methanotrophy.” My issue remains that they've sampled the bacteria from the water column, which is the direct evidence for the presence of these methanotrophic species. In the provided Illumina data, the known methylotrophic species were conspicuously absent. In most cases these species were identified in the Illumina data the ~0.1% range, including the Methylococcales that were highlighted by the authors. This population did vary up to 6%, but only in a couple of samples. Is this a bloom that follows rain events? Could this explain the shrimp diet – do they go on a mad feeding spree when rainwater washes a bulk methane-derived population into the caves? Rather, the provided Illumina data support the idea that the availability of methane as 1% of the organic carbon pool supporting approximately 1% of the microbial community.

The authors state that (as highlighted in table 1), that “methane is a small component of the total organic pool (~1% of the MFW)” – I can see this. So how are the shrimp getting 55% of their diet from the organisms that consume this methane? The newly provided Supplementary Table 5 indicates that the vast majority of microorganisms detected in the water column are heterotrophs – species within the Beta- and Gamma-proteobacteria. While I agree that the humic acid pool is not easily broken down by bacterial species, the Burkholderia and Pseudomonadales are specifically known for their ability to break down complex cyclic, poly-cyclic and aromatic compounds – which constitutes the bulk chemistry of humic (and fulvic) acids. The Pseudomonadales are also ideally suited to growth in anaerobic conditions, where nitrate can serve as an electron acceptor. The argument against nitrate reduction unconvincing; the dominance of the Pseudomonadales would deplete available nitrate within the anoxic zone (indeed, it is one of the metabolic tests used to identify these organisms in the laboratory).

All of this didn't seem to make sense in regard to the quinone pool, which suggested a high relative abundance of quinones associated with methylotrophs. I must admit, this is not my specialty, so I looked at the papers the authors cited. The 1981 Collins and Jones paper does not discuss quinones in methylotrophs, and the 2016 Elling paper discusses archaea, which are substantively absent in the Illumina data. The 1985 Collins and Green paper includes a number of methane-oxidizing species, including *Micrococcus luteus*, which is a happy little heterotroph. This organism can also use C-1 compounds for growth, but if there is more reduced organic carbon present, it will preferentially use that. Could this artificially inflate the quinone pool? Finally, during fermentative growth there is no need for an electron transport chain/quinone pool. Could it be that the dominance of methylotrophic in the quinone pool may simply be due to an under-representation of heterotrophic species due to fermentative processes?

I'm also not sure about the species the authors believe are involved in chemoautotrophy? I didn't see anything significant, other than from the Euryarchaeota and Thaumarchaeota (could the authors separate the Archaea from the Bacteria - lumping them together like this is really confusing). Am I missing something?

In all, I would agree that there is a methane-sink here, but as the authors state, it only represents ~1% of the total carbon pool. If the authors re-wrote the paper to tone down the idea that methanotrophy is the primary driver of animal growth in the aquifer, but that the system provides a previously un-recognized methane sink, that would work well. The can also allude to the question of how the shrimp have become so enriched in the isotopes that suggest a methanotroph-rich diet, that better fits the provided data. I would also remove the idea that microbial community that supports chemoautotrophy – I don't see any evidence of this in the Illumina data. The argument that the methanogenesis occurs in the vadose zone is more compelling, which would argue against it happening in the meteoric freshwater zone (which the wording of the abstract seems to suggest).

Reviewer #2 (Remarks to the Author):

Having read the revised manuscript as well as all reviewer comments and replies, I think that this paper is ready for publication. The authors have sufficiently addressed all comments, including those from reviewer one whom was the most critical. In addition, I do use most of the techniques presented in the MS and they are capable of demonstrating what the authors say they do, as there was some concern about that by reviewer number 1. I do have a suggested modification to a sentence (near the end) for clarity:

Original: The presence of fatty acids that could only originate from microbes in somatic tissues of filter-feeding pelagic-dwelling shrimp...

Suggestion: The presence of fatty acids in somatic tissues of filter-feeding pelagic-dwelling shrimp that could only originate from microbes is strong evidence....

Concern #1: There is an apparent inconsistency between the stable carbon isotope evidence for crustacean tissue containing up to 55% methane carbon, and a) the amount of methane in the cave being small relative to bulk DOC and b) that methanotrophs represent a small fraction of the 16S rRNA sequences.

Response: *This observation is one that has been observed and carefully considered by stable isotope ecologists investigating methane-impacted ecosystems (See Grey, 2016 and references therein). The primary reason methanotrophy is able to contribute such a large fraction of carbon to the food web even as it is not the predominant carbon resource is that virtually all of the methane in the system is oxidized and up to 50% of methane that is oxidized is converted to microbial biomass (Trimmer et al., 2015). By contrast, much of the remaining DOC pool is inaccessible to microbial decomposition, and the fraction that is oxidized is not assimilated with the same efficiency. For example, other heterotrophs assimilate 10-30% of the carbon they oxidize (Trimmer et al., 2015), and chemoautotrophs (e.g., nitrifiers) have efficiencies of ~10% (Feliatra and Bianchi, 1993) or less (Veuger et al., 2013). As stated in the manuscript, the deuterium isotopic data from the study is additional and direct evidence that methane cascades the food web. Methanogenesis causes extreme (and therefore distinct) isotopic fractionation (Whiticar, 1999) that is reflected in the bulk dual isotopic data from shrimp tissue presented in **Fig. 5A**.*

The methane carbon contributions we report for the cave crustaceans are in line with those for higher-order consumers from other aquatic ecosystems (e.g., Bastviken et al., 2003; DelVecchia et al., 2016), especially meromictic (permanently stratified) lakes (e.g. Bastviken et al., 2008; Morana et al., 2015), which are the subaerial analog of the highly stratified subterranean aquifers we are investigating. Nevertheless, in recognition that our statement, "Methanotrophic bacteria comprise up to 55% of the shrimp's diet," distracted the Reviewer 1, we restated the sentence using the average value (21%) from the study. The range of methane contributions among the 27 specimens

we examined was from 0% to 55% (see **Supplementary Table 7** where we also added the average value).

*In response to the concern that the observed microbiology does not support the observation of a strong methane contribution to the food web, we recognize that we did not properly articulate that the samples collected did not target the hot-spots of methane oxidation. Since the initiation of this study we have learned methane oxidation in the caves occurs within a less than 10 cm vertical interval between the anoxic-fresh and hypoxic-brackish water masses (Brankovits, in prep.). The samples collected for this study were bulk samples collected within the anoxic and oxic water masses (not the interface between them). Therefore, the functional groupings presented in **Fig. 4A** and the sequence data in **Supplementary Table 5** are qualitative representation of the microbial taxa present in the water mass. We have clarified the nature of the samples and data in the text to help others avoid the confusion experienced by Reviewer 1 (lines 163 – 165; 403 – 406).*

Concern #2: Our interpretation of the quinone data is called to question because fermentative bacteria lack quinones, and, therefore, the abundance of methylotrophs in the quinone pool may be “inflated.”

Response: *We agree that fermentative bacteria do not use quinones to transport electrons across their membranes. However, that point is not relevant to our discussion because we state in the manuscript that the zone of methanogenesis (where fermentation is also occurring) was not sampled because is located within the saturated portion of the overlying karst bedrock. We state:*

“Limited but detectable abundance of 16S rRNA gene sequences from methanogenic archaea in the MFW suggest that methane production takes place elsewhere. This further supports the hypothesis that the saturated zone within the permeable rock matrix, a portion of the aquifer that was inaccessible to divers, was the most likely source of DOC in the MFW”

Current research on quinones (a nascent field that co-author Becker pioneered during his PhD research) supports that the methylene-ubiquinone, MQ8:7, is associated with aerobic methane oxidation in the environment and MQ8:7 has only been recovered from bacteria that oxidize methane in cultures. We repeat (as discussed above) that the bulk water samples collected for microbiological and lipid analyses were not collected in a manner that would be expected to represent the “hot-spot” of methanotrophic activity. Therefore, the relative abundances for methanotrophs we have reported for the quinones and 16S rRNA amplicon sequences most likely underrepresent their actual abundances and ecological relevance. We do not agree that the proportion of methylotrophs represented in the quinone pool is “inflated,” as suggested by Reviewer 1. To clarify some of our statements about quinones, we moved two quinone references to more appropriate locations in the text (lines 178 – 180), and added an additional reference (line 292, reference #49).

Concern #3: We did not specify what microbes are involved in chemoautotrophy.

Response: *Based on geochemical and stable isotopic evidence, we conclude the primary chemoautotrophic process supporting the food web is related to methanogenesis, which occurs within a portion of the aquifer we did not sample. Specifying the taxonomy of those microbes is therefore not possible, although we have noted the presence of 16S rRNA gene sequences from methanogenic archaea in the methane-rich fresh water layer. We recognize and discuss that other chemoautotrophs may be active, but paired geochemical-sequence sample data that would allow us*

to specify the taxonomy of the chemoautotrophic microbes where the autotrophic processes are occurring was beyond the scope of this study. Again, our sampling protocol did not permit us to target the hot-spots where we hypothesize the key biogeochemical processes are occurring. To more easily distinguish the constituent microbes, we now list the Bacterial and Archaeal domains and their relative abundances separately in **Supplementary Table 5**.

We are hopeful the explanations provided above and our edits to the manuscript address the concerns raised by Reviewer 1 to your satisfaction and provide additional support that methane as a carbon source and methanotrophy as a process are essential components of the tropical subterranean estuary ecosystem we investigated.

Sincerely,

The Authors

References

Bastviken, D., Cole, J.J., Pace, M.L. and Van de Bogert, M.C. (2008) Fates of methane from different lake habitats: Connecting whole-lake budgets and CH₄ emissions. *Journal of Geophysical Research: Biogeosciences* 113.

Bastviken, D., Ejlertsson, J., Sundh, I. and Tranvik, L. (2003) Methane as a source of carbon and energy for lake pelagic food webs. *Ecology* 84, 969-981.

Brankovits, D. (*in prep.*) Hydrological and biogeochemical controls of methane-dynamics and ecosystem function in a tropical subterranean karst estuary, Department of Marine Biology. Texas A&M University at Galveston, College Station.

DeVecchia, A.G., Stanford, J.A. and Xu, X. (2016) Ancient and methane-derived carbon subsidizes contemporary food webs. *Nature Communications* 7.

Feliatra, F. and Bianchi, M. (1993) Rates of nitrification and carbon uptake in the Rhone River plume (northwestern Mediterranean Sea). *Microbial Ecology* 26, 21-28.

Grey, J. (2016) The incredible lightness of being methane-fuelled: stable isotopes reveal alternative energy pathways in aquatic ecosystems and beyond. *Frontiers in Ecology and Evolution* 4, 8.

Morana, C., Borges, A., Roland, F., Darchambeau, F., Descy, J.-P. and Bouillon, S. (2015) Methanotrophy within the water column of a large meromictic tropical lake (Lake Kivu, East Africa). *Biogeosciences* 12, 2077-2088.

Trimmer, M., Shelley, F.C., Purdy, K.J., Maanoja, S.T., Chronopoulou, P.-M. and Grey, J. (2015) Riverbed methanotrophy sustained by high carbon conversion efficiency. *The ISME Journal* 9, 2304.

Veuger, B., Pitcher, A., Schouten, S., Sinninghe Damsté, J.S. and Middelburg, J.J. (2013) Nitrification and growth of autotrophic nitrifying bacteria and Thaumarchaeota in the coastal North Sea. *Biogeosciences* 10, 1775-1785.

Whiticar, M.J. (1999) Carbon and hydrogen isotope systematics of bacterial formation and oxidation of methane. *Chemical Geology* 161, 291-314.

REVIEWERS' COMMENTS:

Reviewer #1 (Remarks to the Author):

I appreciate the authors taking the time to address many of the issues I raised and changing the language of the manuscript. In particular, the manuscript has dialed-back on the original hypothesis that the major contribution to the microbial-loop was methane, and I appreciate how the potential contribution of DOC has now been emphasized.

I still have some concerns as to the authors interpretation of the microbiology in the rebuttal (for example, the Trimmer 2015 paper does not support many of the numbers they pull from it, and I worry about statements such as chemoautotrophs ONLY assimilate ~10% the DOC they oxidize); however, the microbiology in the manuscript itself is sound as written and now appropriately worded regarding the role of the methanogens/trophs.

There are some minor typos:

Ln 298: 16s should be 16S

Ln 301: 'The dominance of type I methanotrophs is not...' should more appropriately be 'The dominance of type I methanotrophs over type II is not...' given that this population is still comparatively small compared to the whole community composition.

Ln 357: *Methylococcaceae* and *Methylocystacea* should be italicized.

Response to Reviewer

Reviewers comments in plain text and *authors' responses italicized*:

Reviewer #1 (Remarks to the Author):

I appreciate the authors taking the time to address many of the issues I raised and changing the language of the manuscript. In particular, the manuscript has dialed-back on the original hypothesis that the major contribution to the microbial-loop was methane, and I appreciate how the potential contribution of DOC has now been emphasized.

I still have some concerns as to the authors interpretation of the microbiology in the rebuttal (for example, the Trimmer 2015 paper does not support many of the numbers they pull from it, and I worry about statements such as chemoautotrophs ONLY assimilate ~10% the DOC they oxidize); however, the microbiology in the manuscript itself is sound as written and now appropriately worded regarding the role of the methanogens/trophs.

***Response:** We appreciate the reviewer's endorsement and we are glad to hear that we satisfied her/his expectations with the revised manuscript. We agree that the statement of concern in the rebuttal is not correct due to the confusing wording on our part. The point we wanted to make is that many chemoautotrophic processes (e.g., nitrification) are inefficient mechanisms for converting inorganic carbon into biomass (e.g., Feliatra and Bianchi, 1993; Veuger et al., 2013).*

There are some minor typos:

Ln 298: 16s should be 16S

Ln 301: 'The dominance of type I methanotrophs is not...' should more appropriately be 'The dominance of type I methanotrophs over type II is not...' given that this population is still comparatively small compared to the whole community composition.

Ln 357: Methylococcaceae and Methylocystacea should be italicized.

***Response:** We have made the required edits in the main text according to Reviewer #1's comments.*

Sincerely,

The Authors

References

- Feliatra, F. and Bianchi, M. (1993) Rates of nitrification and carbon uptake in the Rhone River plume (northwestern Mediterranean Sea). *Microbial Ecology* 26, 21-28.
- Veuger, B., Pitcher, A., Schouten, S., Sinninghe Damsté, J.S. and Middelburg, J.J. (2013) Nitrification and growth of autotrophic nitrifying bacteria and Thaumarchaeota in the coastal North Sea. *Biogeosciences* 10, 1775-1785.